# Computational prediction of CTCF/cohesin-based intra-TAD loops that insulate chromatin contacts and gene expression in mouse liver

Bryan J Matthews*, David J Waxman*

Department of Biology and Bioinformatics Program, Boston University, Boston, United States

**Abstract** CTCF and cohesin are key drivers of 3D-nuclear organization, anchoring the megabase-scale Topologically Associating Domains (TADs) that segment the genome. Here, we present and validate a computational method to predict cohesin-and-CTCF binding sites that form intra-TAD DNA loops. The intra-TAD loop anchors identified are structurally indistinguishable from TAD anchors regarding binding partners, sequence conservation, and resistance to cohesin knockdown; further, the intra-TAD loops retain key functional features of TADs, including chromatin contact insulation, blockage of repressive histone mark spread, and ubiquity across tissues. We propose that intra-TAD loops form by the same loop extrusion mechanism as the larger TAD loops, and that their shorter length enables finer regulatory control in restricting enhancer-promoter interactions, which enables selective, high-level expression of gene targets of super-enhancers and genes located within repressive nuclear compartments. These findings elucidate the role of intra-TAD cohesin-and-CTCF binding in nuclear organization associated with widespread insulation of distal enhancer activity.

DOI: https://doi.org/10.7554/eLife.34077.001

*For correspondence:
bryan.mtthws@gmail.com (BJM);
djw@bu.edu (DJW)

Competing interests: The authors declare that no competing interests exist.

Reviewing editor: Noam Kaplan,

## Introduction

The mammalian genome is organized into stereotypical domains, averaging ~700 kb in length, called Topologically Associating Domains (TADs) (*Dixon et al., 2012*; *Nora et al., 2012*). TADs are insulated chromatin domains whose genomic boundaries are often retained across tissues (*Dixon et al., 2012*) and have been conserved during mammalian evolution (*Vietri Rudan et al., 2015*; *Dixon et al., 2015*). TADs provide a stable genomic architecture that constrains enhancer-promoter contacts, while allowing for dynamic tissue-specific interactions that stimulate gene expression within TADs, thereby linking chromatin structure and positioning to gene expression (*Dowen et al., 2014*; *Sexton et al., 2007*).

Hi-C, an unbiased genome-wide chromosome conformation capture method (*Lieberman-Aiden et al., 2009*), identifies TADs based on their insulation from inter-domain interactions and by the increased frequency of intra-domain interactions that occurs within individual TADs (*Dixon et al., 2012*; *Nora et al., 2012*). TADs show substantial overlap with features of nuclear organization identified using other approaches, including replication domains, lamina-associated domains, and A/B chromatin compartments (*Dixon et al., 2015*; *Pope et al., 2014*; *Nora et al., 2013*). TADs impact gene expression by insulation, which limits a given gene's access to regulatory regions (*Le Dily et al., 2014*). While TAD structures are often shared across tissues within a species, some individual TADs show tissue-specific differences in their spatial positioning within the nucleus, and in their overall activity, transcription factor (TF) binding patterns, and patterns of expression of individual genes

**eLife digest** The human genome contains the complete set of DNA instructions – including all genes – needed to build and maintain an organism. To fit all of this genetic information in the cell's nucleus, the DNA is neatly wrapped around so-called histone proteins, which help to package the genetic material into chromatin, which forms thread-like structures, the chromosomes.

Chromatin is further folded into large DNA loops held together by an anchor protein, CTCF, and by a second protein, cohesin, whose ring-shaped structure ties each loop at its base. DNA segments that are within the same loop may interact frequently, whereas those outside the loop rarely do. Many of these large DNA loops are further pinched off into sub-loops. These sub-loops may help a cell fine-tune whether a gene needs to be turned on or off by limiting the contact between genes and the DNA regions that regulate the activity of genes.

Knowing where these DNA sub-loop are located is very important for understanding how each gene is controlled. However, this can be very costly to determine, and therefore, is only known for a few cell types. Now, Matthews and Waxman tackle this issue by creating a computer model that can correctly predict many of these sub-loops. The method used experimental data obtained from mouse liver cells to identify the locations of CTCF and cohesin.

The results showed that DNA sub-loops in the liver cells can shield genes from regulatory DNA segments outside the looped area. For example, a small sub-loop that contains a single gene related to obesity is highly active, even though the large DNA loop containing the sub-loop is an otherwise inactive gene region. Similarly, certain genes critical for liver function are positioned within sub-loops containing DNA regions that greatly enhance the gene activity in liver cells. This allows the selected genes to be highly active – unlike other genes that are close by but outside the sub-loop.

This new approach will make it easier and cheaper to discover DNA loops and sub-loops across the genome. A better knowledge of where these loops form may also allow us to better understand how genes are turned on and off in different types of cells, and in response to biological stimuli or environmental stresses. This may also help understand and treat conditions that arise from mutations that disrupt the boundaries of DNA loops or sub-loops, which can allow certain DNA segments to activate the wrong genes and can lead to developmental defects and diseases such as cancer.

DOI: https://doi.org/10.7554/eLife.34077.002

---

(*Dixon et al., 2015*). It is unclear to what extent these large megabase-scale chromatin structures exert regulatory control over the multiple, often variably-expressed, genes found within their boundaries.

Two key protein factors, CCCTC-binding factor (CTCF) and the multi-subunit cohesin complex, are the primary architects of nuclear organization in mammals (*Ong and Corces, 2014*; *Sanborn et al., 2015*; *Guo et al., 2015*). CTCF and cohesin cooperatively engage genomic DNA via a loop extrusion complex, which is dynamically mobile within TAD boundaries and may help organize TAD structure (*Sanborn et al., 2015*; *Fudenberg et al., 2016*; *Rao et al., 2014*). CTCF is an 11 zinc finger protein that stably binds DNA and can serve as an insulating enhancer-blocker and a modulator of 3D chromatin structure (*Phillips and Corces, 2009*). Sites bound by both cohesin and CTCF (cohesin-and-CTCF (CAC) sites) are associated with insulator function (*Dowen et al., 2014*; *Zuin et al., 2014*) and are found at TAD boundaries (*Dixon et al., 2012*; *Nora et al., 2012*). In contrast, sites bound by cohesin but not CTCF (cohesin-non-CTCF (CNC) sites) are found at tissue-specific promoters and enhancers (*Kagey et al., 2010*) and may help to stabilize large TF complexes (*Faure et al., 2012*). CAC complexes are also associated with topoisomerase-IIβ (Top2b), which presumably relieves the torsional strain of the extrusion complex (*Uusküla-Reimand et al., 2016*).

Complete knockout of either CTCF or cohesin is embryonic lethal (*Heath et al., 2008*; *White et al., 2013*; *Xu et al., 2010*), whereas partial depletion of CTCF or cohesin results in altered gene expression but has more limited phenotypic impact, increasing radiation sensitivity, DNA repair defects, and cell cycle arrest (*Ong and Corces, 2014*; *Xu et al., 2010*; *Moore et al., 2012*). Complete removal of CTCF or cohesin-related factors, achieved using inducible degradation systems,

leads to a complete loss of virtually all loop structures in a highly dosage-dependent manner (*Nora et al., 2017*; *Rao et al., 2017*; *Schwarzer et al., 2017*). Mutations affecting CAC loop anchors are frequently seen in cancer and lead to dysregulation of adjacent genes, evidencing the functionality of these loops (*Ji et al., 2016*; *Katainen et al., 2015*; *Fujimoto et al., 2016*). However, there are many more CAC sites within TADs than at TAD boundaries, and it is not clear what factors differentiate loop-forming CAC sites at TAD boundaries from other CAC sites in the genome.

Chromatin interactions can be studied by Hi-C analysis, which under standard conditions provides a resolution of 25–100 kb and has been used to study nuclear organization at the level of megabase-scale TAD structures. However, high resolution Hi-C datasets obtained using extreme deep sequencing (>25 billion reads) have led to two key discoveries (*Rao et al., 2014*). First, ~90% of DNA loops ('loop domains', defined as local peaks in the Hi-C contact matrix) are associated with both CTCF binding and cohesin binding, and 92% of such loops involve inwardly oriented CTCF anchors (*Rao et al., 2014*). Thus, loop anchors are bound at asymmetric CTCF motifs that face the loop interior. This previously unappreciated feature of CTCF loops facilitates the identification of such loops *in silico* (*Sanborn et al., 2015*; *Oti et al., 2016*). Furthermore, expression of neighboring genes changes in a predictable manner when CTCF anchors are inverted or deleted by CRISPR/Cas9 genomic editing (*Dowen et al., 2014*; *Sanborn et al., 2015*; *Guo et al., 2015*). Second, extreme deep sequencing Hi-C studies identify a much larger number of shorter loops than previously recognized (~10,000 loops with a median size of 185 kb) (*Rao et al., 2014*), many of which represent complex nested structures (e.g., isolated cliques) (*Sanborn et al., 2015*). The ability to distinguish between such substructures has led to predictions ranging from $10^3$ to $10^6$ loops per genome, depending on the 3C-based analysis method and the cutoff values employed (*Sanborn et al., 2015*; *Handoko et al., 2011*; *Fullwood et al., 2009*; *Jin et al., 2013*). The presence of nested loop structures may be a general feature of topological nuclear organization, and the ability to detect such structures is dependent on the method, resolution, and computational approach (*Handoko et al., 2011*; *Fullwood et al., 2009*; *Jin et al., 2013*; *Hnisz et al., 2016*; *Weinreb and Raphael, 2016*).

While sub topologies within TADs have been observed, it is unknown whether those interactions represent enhancer-promoter loops or other looped structures, and whether they are mediated by cohesin, mediator, or other architectural proteins (*Zuin et al., 2014*; *Sofueva et al., 2013*). Short, <200 kb CTCF-anchored loops, termed chromatin contact domains or super-enhancer domains, have been identified in mouse embryonic stem cells (mESCs) by ChIA-PET experiments that select for CTCF and cohesin binding sites (via immunoprecipitation of Smc1) (*Handoko et al., 2011*; *Tang et al., 2015*), and are enriched for tissue-specific genes and enhancers (*Dowen et al., 2014*; *Handoko et al., 2011*). However, these genomic regions represent a minority of CTCF-anchored DNA loops, and likely do not fully represent all of the nuclear topological domains evident in high resolution Hi-C maps (*Rao et al., 2014*; *Rao et al., 2017*; *Rowley et al., 2017*). Given the inability to identify CAC-anchored intra-TAD loops from standard, low resolution Hi-C data, we sought to build on the above advances and develop a computational method to predict such subTAD-scale loops by using only 2D (CTCF and cohesin ChIP-seq binding activity) and 1D (CTCF motif orientation) information. Here we define intra-TAD loops anchored by cohesin and CTCF, and that contain at least one gene, which represent a superset encompassing super-enhancer and polycomb domains (*Dowen et al., 2014*). These CAC-mediated intra-TAD loops are mechanistically distinct from short range enhancer-promoter loops, and from longer range genomic compartmentalization (*Rao et al., 2017*; *Schwarzer et al., 2017*; *Stevens et al., 2017*), whose impact on gene expression in mouse liver is also discussed.

Here we present, and then validate in three mouse tissues and two human cell lines, a computational method to identify intra-TAD loops genome-wide. We elucidate the structural and functional features of the intra-TAD loops identified, and those of the better-established TADs, including their impact on gene expression in a mouse liver model. We show that, mechanistically, intra-TAD loops are anchored by loop extrusion CAC complexes that are shared across tissues and show strong conservation. Further, we demonstrate that, at a functional level, intra-TAD loops insulate repressive chromatin mark spread and thereby enable selective expression of genes at a high level compared to their immediate genomic neighbors, notably genes targeted by super-enhancers, and genes that are otherwise found in repressive nuclear compartments. These findings reveal how intra-TAD loops harness many of the same mechanisms as TAD-scale loops but in ways that allow for greater local control of gene expression.

## Results

### Features of TADs and their functional impact on gene expression

#### Features associated with TAD boundaries

We characterized TADs identified in mouse liver (*Vietri Rudan et al., 2015*) using matched ChIP-seq datasets for CTCF and the cohesin subunit Rad21, which we obtained for a group of individual adult male mouse livers. Genomic regions co-bound by cohesin and CTCF (CAC sites) were strongly enriched at TAD boundaries (*Figure 1A*), consistent with (*Vietri Rudan et al., 2015*). In contrast, cohesin-non-CTCF sites (CNC sites) were weakly depleted at TAD boundaries. We also observed strong enrichment for motif-oriented CTCF binding at TAD boundaries (*Figure 1—figure supplement 1A*), consistent with recent reports and the loop extrusion model of domain formation (*Vietri Rudan et al., 2015*; *Sanborn et al., 2015*; *Rao et al., 2014*). Next, we explored the impact of cohesin depletion on CAC sites associated with TAD boundaries, following up on the finding that many cohesin binding sites are maintained upon knockout or knockdown of components of the cohesin complex (*Faure et al., 2012*). *Figure 1B* shows the distribution of cohesin-bound regions that are either resistant or sensitive to haploinsufficiency of the cohesin subunit Rad21 in hepatocytes, or to knockout of the cohesin subunit Stag1 in mouse embryonic fibroblasts (MEFs). In both cell types, sites resistant to cohesin loss are enriched at TAD boundaries, while those sensitive to cohesin loss are more equally distributed along the TAD length. This may explain the unexpected finding that domains and compartments are largely maintained after depletion of cohesin (*Zuin et al., 2014*; *Sofueva et al., 2013*; *Seitan et al., 2013*).

Given the frequent conservation of TAD boundaries between tissues in both mouse and human (*Dixon et al., 2012*; *Nora et al., 2012*; *Dixon et al., 2015*), we compared regions bound by CTCF in mouse liver to 15 other mouse tissues from the ENCODE Project (*Shen et al., 2012*). CTCF sites that are shared across 12 or more tissues showed 3–4 fold enrichment at TAD boundaries relative to the center of the TAD, whereas CTCF binding sites unique to liver, or shared with only one other tissue, showed no such enrichment (*Figure 1C*). TAD boundaries were also enriched for CpG hypomethylation, which was most pronounced at TAD anchor CTCF motifs (*Figure 1D*, *Figure 1—figure supplement 2A*). CpG methylation is greater at CAC sites not involved in TAD or intra-TAD loop anchors (*Figure 1—figure supplement 2A*; see below), and could represent an additional layer of epigenetic regulation of CAC-based loop formation. By comparison, the TAD boundary enrichment seen for CTCF and cohesin was absent for >50 other liver-expressed TFs whose binding site distribution within TADs we examined (*Figure 1E*, and data not shown). Tbp and E2f4, which are characterized by promoter-centric binding (*Blanchette et al., 2006*; *Kim et al., 2005*), are two notable exceptions (*Figure 1—figure supplement 1B*). Consistent with this, TAD boundaries were enriched for promoters of protein-coding genes, including promoters that do not overlap CAC sites, and for histone marks associated with promoters but not enhancers *Figure 1—figure supplement 1C–E*).

#### TADs segregate the genome into compartmentalized units

TADs have the ability to insulate the spread of repressive histone marks and also enhancer-promoter interactions, referred to as enhancer blocking (*Dixon et al., 2012*; *Dowen et al., 2014*; *Sofueva et al., 2013*). By these dual mechanisms, TADs can exert control over tissue-specific gene expression, despite the TADs themselves being largely structurally invariant across tissues. As TADs are defined based on their insulation of chromatin contacts, we investigated their impact on chromatin mark spread. We examined four broad histone marks associated with either transcriptional repression (H3K9me3, H3K27me3) or activation (H2AK5ac, H3K36me3). We also examined Global Run-on Sequencing data to identify actively transcribed regions of the genome, as well as Lamina Associated Domain (LAD) coordinates to visualize areas of the genome associated with the nuclear periphery. *Figure 1F* shows a heat map representation of a 1 Mb window around each TAD boundary in mouse liver, clustered using k-means clustering (k = 4) based on H3K9me3 and H2AK5ac ChIP-seq data and on the Eigen value of the Hi-C principal component analysis (PCA), which provides an estimate of active versus inactive genomic compartments (*Lieberman-Aiden et al., 2009*). A subset comprised of 1439 liver TAD boundaries (40.9% of all boundaries) represents transitions from inactive to active chromatin compartments, or vice versa (*Figure 1F*; 2nd and 3rd clusters). Also shown is an example of a transitional TAD boundary on chromosome 13, where there is a shift from

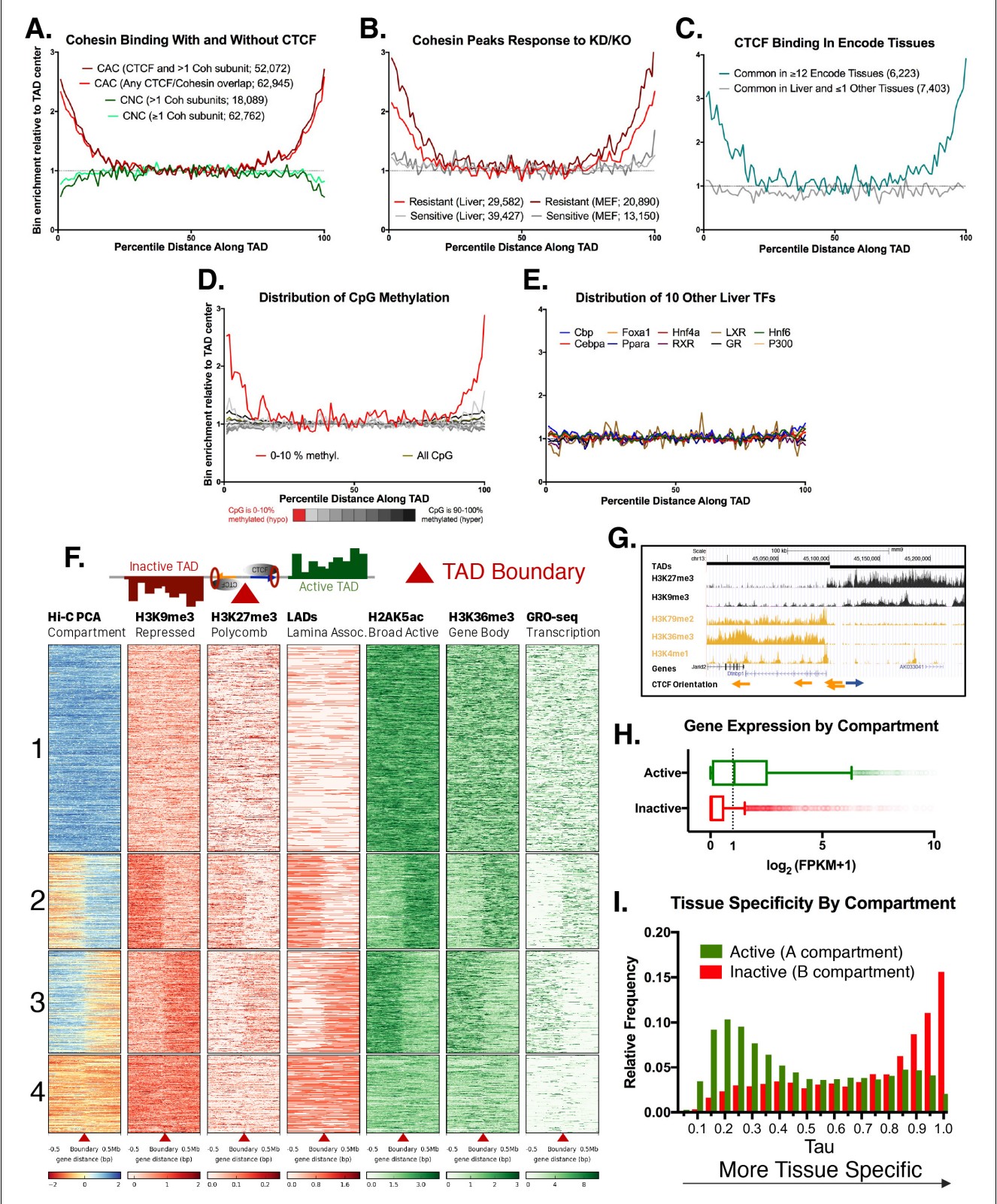

**Figure 1.** Features of TAD boundaries and TAD insulator function. Profiles in A-E represent a normalized aggregate count of peaks or features along the length of all TADs, sub-divided into 100 equally-sized bins per TAD, where bin #1 is the 5′ start of the TAD and bin #100 is at the TAD 3′ end. Normalization was performed to allow comparison of multiple groups with variable peak numbers in a single figure. The y-axis displays the enrichment within a given bin versus the average of the five center bins (bins #48–52). In A-C, the number of binding sites in each group is shown in parenthesis. (A)

*Figure 1 continued*

Cohesin-and-CTCF (CAC) sites are enriched at TAD boundaries, while cohesin-non-CTCF (CNC) sites are weakly depleted. As the cohesin (Coh) complex is a multi-protein complex, the darker color within each group represents a stricter overlap between cohesin subunits (Rad21, Stag1 and Stag2). (B) In both mouse liver and MEFs, cohesin binding sites that are resistant to knockdown (KD) or knockout (KO) of cohesin component subunits (~40% of cohesin binding sites for liver) are strongly enriched for TAD boundaries. Cohesin sites that are sensitive to loss following KD or KO (~60% of sites for liver) are not enriched at TAD boundaries. (C) CTCF binding sites in liver that are deeply-shared across other ENCODE tissues (≥12 out of 15 other tissues examined) are strongly enriched at liver TAD boundaries, while those that are either unique to liver or shared in only one other tissue are not enriched at TAD boundaries. (D) TAD boundaries show greater hypomethylation than the TAD interior. The most hypomethylated CpGs are enriched at TAD boundaries, which likely represents a combination of hypomethylation at gene promoters and hypomethylation at CTCF binding sites. CpG methylation states, determined by liver whole genome bisulfite sequence analysis were subdivided into 10 bins based on the degree of methylation (0–10% methylated, 10–20%, etc.) prior to TAD distribution analysis. (E) 10 liver-expressed TFs are not enriched at TAD boundaries. These profiles are representative of the vast majority of the >50 publically available ChIP peak lists for liver-expressed TFs. Notable exceptions, related to promoter-associated features, marks, and transcription factors, are shown in *Figure 1—figure supplement 1B,D*. (F) Shown is a heat map of the distribution of the indicated activating and repressive marks and other features determined for male mouse liver across a 1 Mb window around each TAD boundary. TAD clusters, numbered at the left, were defined using k means clustering (k = 4). The boundaries between TADs transition from active to inactive chromatin compartments (or vice versa) for TAD clusters 2 and 3. In downstream analyses based on these results, a TAD was considered active if the boundary at the start of a TAD fell into clusters 1 or 2 and the boundary at the end of the same TAD fell into clusters 1 or 3; inactive TADs are those whose boundaries begin in clusters 3 or 4 and end in clusters 2 or 4 (see Materials and methods). See *Supplementary file 1A* for a full listing of the 3538 autosomal TADs analyzed and their active/inactive status. (G) UCSC browser screenshot for a transitional TAD boundary on chromosome 13 from TAD cluster 3 in *Figure 1F*. Arrows at bottom indicate CTCF motif orientation. (H) Box plots showing liver gene expression (RNA-seq) for 12,258 genes in 1930 active TADs and 4643 genes in 1000 inactive TADs (*Supplementary file 1A*). 939 genes in 473 of the inactive TADs are expressed at >1 FPKM (*Supplementary file 1E*). Genes found in active compartment TADs are more highly expressed, with the majority of genes showing >1 FPKM, than genes found in inactive TAD compartments. Genes in weakly active and weakly inactive TADs were excluded from these analyses. (I) Genes whose TSS are located in inactive TADs ('B compartments') are more tissue specific in their expression pattern than genes found in active TADs ('A compartments'). The top GO category for expressed genes in the A compartment is RNA binding, while the top category for expressed genes in the B compartment is monooxygenase activity (not shown).

DOI: https://doi.org/10.7554/eLife.34077.003

The following figure supplements are available for figure 1:

**Figure supplement 1.** Additional Features of TAD boundaries.
DOI: https://doi.org/10.7554/eLife.34077.004
**Figure supplement 2.** Additional Features of TAD boundaries, continued.
DOI: https://doi.org/10.7554/eLife.34077.005

active to inactive chromatin marks separated by inversely-oriented CTCF binding sites (*Figure 1G*). Using the clusters shown in *Figure 1F*, each TAD was designated as active, weakly active, inactive, or weakly inactive, based on the signal distribution around the boundary and the eigenvalue from Hi-C PCA analysis along the length of the TAD (see Materials and methods and listing of TADs in *Supplementary file 1A*). Striking differences in gene expression were seen between active and inactive compartment TADs (median liver expression 1.095 FPKM for 12,258 genes in active TADs vs. 0.003 FPKM for 4643 genes in inactive TADs; *Figure 1H*).

We sought to determine the tissue-specificity of the genes in active vs. inactive TADs. We used the expression level of each gene across ENCODE tissues to calculate Tau scores, a robust metric for tissue specificity (*Yanai et al., 2005*; *Kryuchkova-Mostacci and Robinson-Rechavi, 2017*). Tau values close to one are highly tissue specific, while lower values (<0.3) are widely expressed and considered housekeeping genes (see Materials and methods). A greater fraction of genes located in inactive TADs are tissue-specific compared to genes in active TADs (*Figure 1I*). Overall, only 939 (20.2%) of all 4643 genes in inactive TADs are expressed in liver (FPKM >1) vs. 6,290 (51.3%) of the 12,258 genes in active TADs. Furthermore, genes whose TSS is close to a TAD boundary (i.e., TSS within 2% of the total TAD length in either direction from the boundary) tend to be less tissue specific than the genomic average (*Figure 1—figure supplement 2B*). Active transcription may be a key driver of dynamic cohesin movement in the nucleus (*Busslinger et al., 2017*), and RNA polymerase II, in vitro, is capable of translocating cohesin rings along DNA (*Davidson et al., 2016*). Thus, the ubiquitous expression of genes at TAD boundaries could be either a driver or an initiator of loop extrusion, although the exact mechanism remains unknown.

## Identification of intra-TAD loops

### Predicting intra-TAD loops from TAD-internal CTCF and cohesin binding sites

While CAC sites that are tissue ubiquitous, cohesin knockdown-resistant, or species-conserved show a clear 2 to 5-fold enrichment at TAD boundaries (*Figure 1*), a large majority of such sites are TAD-internal and presumably do not contribute to TAD formation. Overall, only 14.7% of liver CTCF binding sites are associated with TAD boundaries (*Figure 2—figure supplement 1A*), consistent with other reports (*Dixon et al., 2012*), and only 23% of the CTCF-bound regions that retain all four of the above features are within 25 kb of a TAD boundary (*Figure 2—figure supplement 1A*). We considered two possibilities: (1) TAD-internal CAC sites form intra-TAD loops that are too short to be detected in standard Hi-C datasets; and (2) additional factors associated with TAD boundary CTCF sites differentiate them from other such binding sites in the genome (see Discussion). To examine the first possibility, we modified an algorithm for analysis of CTCF loops (*Oti et al., 2016*) and adapted it to predict subTAD-scale loops in silico, using CTCF and cohesin peak strength and CTCF orientation as inputs (*Figure 2A*). Our approach builds on the finding that >90% of CTCF-based loops are formed between inwardly-oriented CTCF sites (*Rao et al., 2014*). Each mouse liver CAC site was given a score that represents its CTCF peak strength and CTCF motif score, and an orientation was assigned based on whether the non-palindromic CTCF motif was present on the (+) strand or the (-) strand, considering the highest scoring CTCF motif at each CAC site. Scanning the genome, each (+) strand CAC peak was connected to putative downstream (-) strand CAC sites. Low scoring CAC peaks were removed and the process was iteratively repeated until the top 20,000 candidate loops remained. The set of loops was then filtered, as detailed in Materials and methods, to take into account cohesin scoring, and to ensure TSS overlap and <80% TAD overlap, to restrict our definition of intra-TAD loops to TAD-internal CAC-mediated loops that contain at least one TSS. Applying this algorithm to each of 4 matched pairs of liver ChIP datasets for CTCF and cohesin, we identified a set of 9543 intra-TAD loops present in all four liver samples, with a median length of 151 kb. The set of intra-TAD loops identified includes many nested loops, and differs substantially from the generally shorter and much larger number of CTCF loops predicted by the original algorithm (*Oti et al., 2016*) (*Figure 2—figure supplement 1B*; see Materials and methods). Functionally, anchors of the shorter loops predicted using the method of (*Oti et al., 2016*) show less insulation and weaker directional interactions than the intra-TAD loop anchors identified in our study (*Figure 2—figure supplement 1C,D*; also see below). Moreover, 91% of our predicted intra-TAD loops were wholly contained within a single TAD, versus only 67% for a random shuffled control (*Figure 2—figure supplement 1E*). *Figure 2B* illustrates intra-TAD loop structures within TADs along a segment of chromosome two and highlights their substantial overlap with 'CTCF-CTCF' DNA loops identified by ChIA-PET analysis of cohesin-mediated interactions in mouse embryonic stem cells (mESCs) using antibodies to the cohesin subunit Smc1 (*Dowen et al., 2014*). The final set of 9543 liver intra-TAD loops includes 1632 intra-TAD loops (17.1%) that share one CTCF loop anchor with a TAD boundary (i.e., an intra-TAD loop nested in a TAD with a potential shared anchor). Consistent with these findings, high resolution Hi-C data in mouse CH12 cells reveals the presence of single, multiple, and more complex nested intra-TAD loops that were predicted in mouse liver (*Figure 2—figure supplement 2A–C*).

### Intra-TAD loop anchors share many properties of TAD anchors

We examined the set of predicted intra-TAD loops and their CAC site anchors to investigate their impact on genome structure and gene regulation. For these analyses, we excluded from the intra-TAD anchor group the 1632 intra-TAD anchors that are shared with TAD anchors to ensure that the groups compared are mutually exclusive (*Figure 2—figure supplement 3A*). We first sought to determine if the intra-TAD loop anchors show conserved CTCF binding across multiple ENCODE tissues, as seen for TADs in *Figure 1C*. *Figure 2C* shows the tissue distribution of CTCF binding at CTCF binding sites found at intra-TAD loop anchors in liver, where an x-axis value of 1 indicates the liver CTCF binding site is occupied by CTCF in only one other tissue, and a value of 15 indicates binding occurs in all 15 mouse tissues where CTCF ChIP-seq data is available. Results show that a large majority of intra-TAD loop anchors are bound by CTCF in at least 10 of the 15 mouse tissues examined. Indeed, CTCF binding at the TAD-internal intra-TAD loop anchors is more deeply

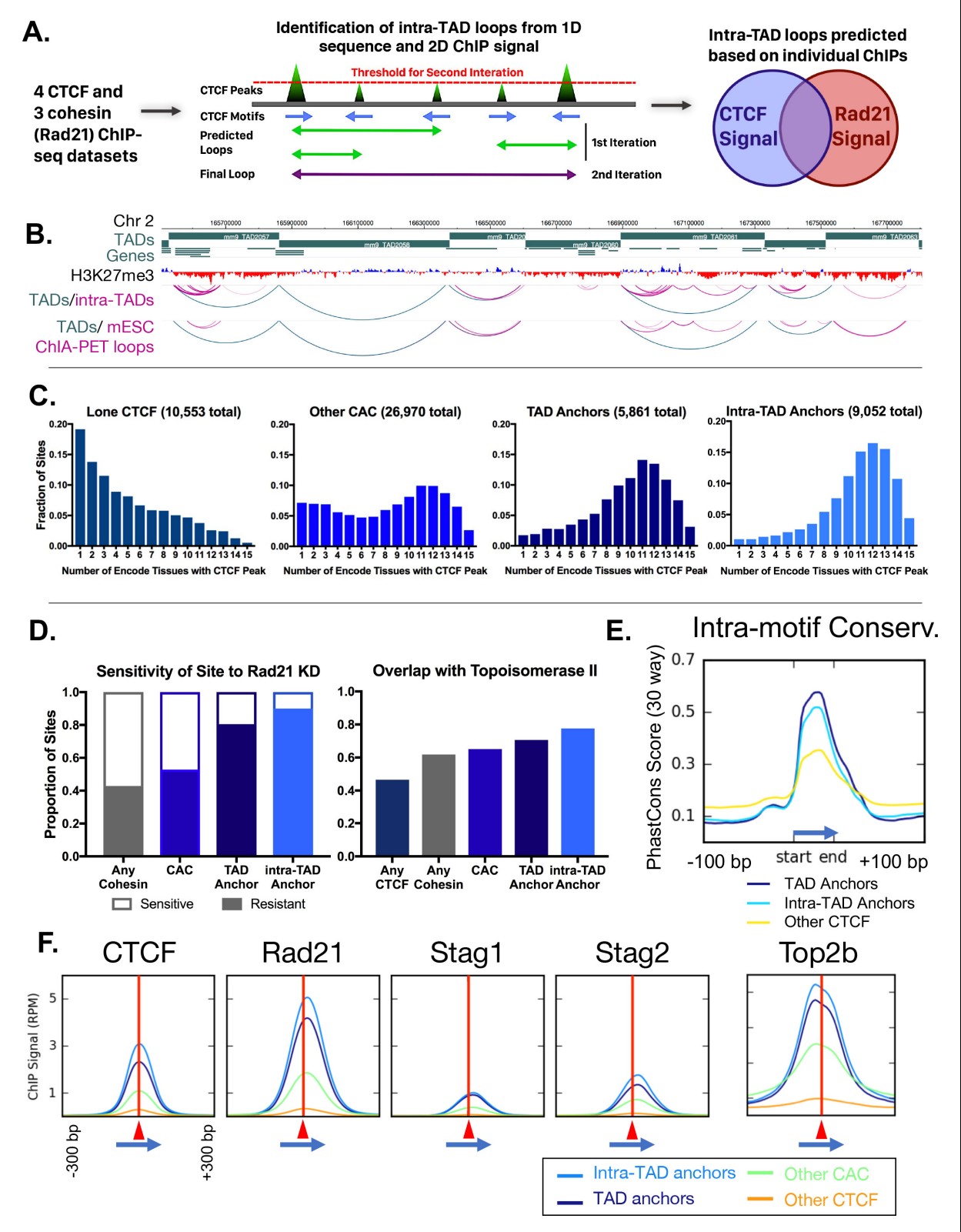

**Figure 2.** Predicted intra-TAD loop anchors share many properties of TAD anchors. (**A**) Diagram illustrating intra-TAD loop prediction based on CTCF motif orientation and CTCF and cohesin (Rad21) ChIP-seq binding strength data. Iteration was conducted until 20,000 loops were predicted per sample, prior to filtering and intersection across samples, as detailed in Materials and methods. (**B**) Shown is a 2 Mb segment of mouse chromosome 2 indicating TAD loops (blue) and intra-TAD loops (pink) in relation to genes. Also shown are cohesin interaction loops identified experimentally in mouse

*Figure 2 continued on next page*

*Figure 2 continued*

ESC by Smc1 ChIA-PET (*Dowen et al., 2014*). (C) TAD and predicted intra-TAD loop anchors are more tissue ubiquitous than other categories of CTCF/CAC sites. Each of the four CTCF site subgroups was defined in mouse liver as detailed in *Supplementary file 1C*. The x-axis indicates the number of ENCODE tissues out of 15 tissues examined that also have CTCF bound, where a higher value indicates more tissue-ubiquitous CTCF binding. These data are shown for 'lone' CTCF binding sites (10,553), non-anchor cohesin-and-CTCF sites ('Other CAC'; 26,970), TAD anchors (5,861), and intra-TAD loop anchors (9,052, which excludes those at a TAD loop anchor). While 'Other CAC' sites tend to be weaker (*Figure 2F*, below), 93% are bound by CTCF in at least one other mouse tissue, and 66% were verified in at least six other tissues. Similarly, for 'Lone CTCF', 81% of sites were bound by CTCF in at least one other mouse tissue, and 39% were verified in at least six other tissues (*not shown*). (D) TAD and intra-TAD loop anchors are more resistant to the knockdown effects of Rad21 ±haploinsufficency than other CAC sites or cohesin-bound regions. A larger fraction is also bound by the novel extrusion complex factor Top2b (*Supplementary file 1C*). (E) Loop anchors show greater intra-motif conservation than other CTCF-bound regions. Shown are the aggregate PhastCons score for oriented core motifs within either TAD (dark blue) or intra-TAD (light blue) anchors as compared to other CTCF peaks with motifs (yellow). (F) Cohesin interacts with the COOH terminus of CTCF (*Xiao et al., 2011*), which resulting in a shift of ~20 nt in cohesin ChIP signal relative to the CTCF summit (c.f. shift to the right of vertical red line) regardless of category of CTCF binding site (anchor/non-anchor). Blue arrows indicate the CTCF motif orientation and red triangles and vertical lines indicate position of the CTCF signal summit.

DOI: https://doi.org/10.7554/eLife.34077.006

The following figure supplements are available for figure 2:

**Figure supplement 1.** Comparison of CTCF Features within TADs and Loop Prediction Improvements.
DOI: https://doi.org/10.7554/eLife.34077.007
**Figure supplement 2.** Example Screenshots of Predicted intra-TAD Loops with Observable Interactions in CH12-LX (Mouse B-Cells).
DOI: https://doi.org/10.7554/eLife.34077.008
**Figure supplement 3.** Subclasses of CTCF binding events in relation to predicted loops.
DOI: https://doi.org/10.7554/eLife.34077.009
**Figure supplement 4.** Intra-TAD loop prediction in two other mouse cell types: mESC and NPC.
DOI: https://doi.org/10.7554/eLife.34077.010
**Figure supplement 5.** Example Screenshots for intra-TAD Loops in mESC and NPC (Mouse).
DOI: https://doi.org/10.7554/eLife.34077.011
**Figure supplement 6.** Intra-TAD loop predictions in human cell lines GM12878 and K562.
DOI: https://doi.org/10.7554/eLife.34077.012
**Figure supplement 7.** Example Screenshots for intra-TAD Loops in K562 and GM12878 (Human).
DOI: https://doi.org/10.7554/eLife.34077.013

conserved across mouse tissues than that at TAD boundaries. In contrast, CTCF sites not associated with cohesin binding (lone CTCF sites), and to a lesser extent CAC sites not at intra-TAD or TAD loop anchors (other CAC sites), showed much greater tissue specificity for CTCF binding (*Figure 2C*).

The enrichment of knockdown-resistant cohesin binding sites at TAD boundaries, seen in *Figure 1B*, may explain the persistence of domain structures following CTCF or cohesin depletion (*Zuin et al., 2014*; *Seitan et al., 2013*). Further, we found that 80% of TAD anchors and 90% of intra-TAD loop anchors are resistant to the loss of cohesin binding in Rad21$^{+/-}$ mice vs. only 52.8% for all CAC sites (*Figure 2D*). Moreover, a large fraction of TAD and intra-TAD loop anchors, 70.6% and 77.6%, respectively, are comprised of 'triple sites', where cohesin and CTCF are co-bound with Top2b, a potential component of the loop extrusion complex (*Uusküla-Reimand et al., 2016*), vs. only 46.6% for the set of all CTCF sites (*Figure 2D*). Top2b binding appears to be associated with cohesin binding rather than with CTCF binding, as it is present at enhancer-like CNC sites much more frequently than at CTCF sites in the absence of cohesin (*Figure 2—figure supplement 3B*).

TAD and intra-TAD loop anchors show greater sequence conservation within the core 18 bp CTCF motif than other CTCF sites (*Figure 2E*). Analysis of sequences surrounding the CTCF core motif did not provide evidence for loop anchor-specific motif usage or cofactor binding (*Figure 2—figure supplement 3C,D*). Downstream from the core CTCF motif (within the loop interior) we observed a shoulder of high sequence conservation, as well as additional complex motif usage, likely due to the multivalency of CTCF-DNA interaction, as described in (*Nakahashi et al., 2013*). TAD and intra-TAD loop anchors showed broader and more complex CTCF motif usage outside of the core (*Figure 2—figure supplement 3C,D*); however, only a small minority of sites contained any specific motif in this flanking region. It is less clear to what extent this broader motif usage is a general property of strongly-bound CTCF regions or of the loop anchors themselves. In fact, we observed a consistent positioning of the cohesin peak just downstream of the CTCF peak,

independent of whether the CTCF site was predicted to participate in loop formation or not (i.e. at both loop anchors and 'Other CAC' sites; *Figure 2F*), in accordance with the loop extrusion model and other observations (*Rao et al., 2014*; *Kagey et al., 2010*; *Faure et al., 2012*; *Xiao et al., 2011*).

## Intra-TAD loops in other mouse tissues and in human cells

Given the highly tissue-conserved binding of CTCF at sites that we predicted to serve as intra-TAD loop anchors in liver (*Figure 2C*), we sought direct experimental evidence for the presence of these loops in two other mouse cell types, ESCs and NPCs, where domain and loop definitions have been established based on high-resolution Hi-C datasets (*Bonev et al., 2017*). Predicted loops were similar in number and size across cell types, with substantial overlap between intra-TAD loops predicted in liver compared to loops identified experimentally by Hi-C in mESCs or NPCs (62–63%; *Figure 2—figure supplement 4A*). We also observed 57–63% overlap of our predicted intra-TAD loops across the 3 cell types with CAC-anchored loops identified in mESC using ChIA-PET for the cohesin subunit Smc1 (*Dowen et al., 2014*; *Handoko et al., 2011*; *Hnisz et al., 2016*) (*Figure 2B*, *Figure 2—figure supplement 4A*).

A majority of the predicted liver intra-TAD loops were also found in newer cohesin HiChIP datasets (ChIP for Smc1a followed by Hi-C; data not shown) (*Mumbach et al., 2016*). The substantial overlap between intra-TAD loops across mouse cell types is very similar to the overlap between TADs, indicating a similar level of tissue ubiquity for intra-TAD loops as for TADs (*Figure 2—figure supplement 4B*). Further, loops that were predicted in multiple cell types had stronger interactions than tissue-specific or other loops, as determined by Smc1 ChIA-PET in mESC (*Figure 2—figure supplement 4C*). A recent pre-print corroborates this result in human cells, where tissue-shared CTCF loops were much stronger than tissue-specific CTCF loops (*Kai et al., 2017*).

Overall, ~75% of intra-TAD loops that we identified in mouse liver are experimentally observed in at least one other cell type (mESCs, CH12, or mouse NPCs). Further, 48.5% (4,632) of the computationally predicted liver intra-TAD loops were also present in both mESCs and NPCs (*Supplementary file 1B*). By comparison, 26.2% and 21.5% of CTCF loops in HeLa and K562 cells, respectively, were tissue-specific, suggesting that the ~25% of loops without support in at least one other tissue likely represent bona fide (albeit weaker) liver-specific intra-TAD loops (*Kai et al., 2017*). Examples of both shared and tissue-specific intra-TAD loops with supporting high resolution Hi-C interactions are shown in *Figure 2—figure supplement 5A–C*.

We also predicted intra-TAD loops for two human cell lines, GM12878 and K562, and then compared our predictions to loop domains and contact domains identified in these cells (*Rao et al., 2014*). We predicted more loops in human cells (~15,000 loops that contain a TSS) than in mouse (~10,000 loops with a TSS), owing in part to the lack of a TAD overlap filter. We observed substantially more overlap of the predicted set of intra-TAD loops with loop domains (40–54%) or with K562 cell CTCF ChIA-PET interactions (60–65%) than with contact domains (26–35%; *Figure 2—figure supplement 6A*). Biologically, this difference makes sense, as ChIA-PET and our intra-TAD prediction method both define a CTCF/CAC mediated interactome. Intra-TAD loops are more commonly shared between K562 and GM12878 cells (67–73% shared) as compared to loop domains (46–66%) or contact domains (37–57%) (*Figure 2—figure supplement 6B*); this can be compared to the much smaller difference in percentage overlap (only 2–6%) between TADs and intra-TAD loops seen in mouse cells (*Figure 2—figure supplement 4B*). As in mouse, intra-TAD loops that were predicted in both K562 and GM12878 cells interacted more strongly than K562-specific or other loops, as determined by CTCF ChIA-PET in K562 cells (*Figure 2—figure supplement 6C*). Further, we found evidence for bona fide tissue-specific and shared intra-TAD loops (*Figure 2—figure supplement 7A–C*), as well as an example of a tissue-specific enhancer-promoter interaction in GM12878 cells within a larger tissue-specific intra-TAD loop (*Figure 2—figure supplement 7B*).

## Intra-TAD loops show strong, directional interactions and insulate chromatin marks

TADs are proposed to impact gene expression via two types of insulation: by insulation of chromatin interactions (also called enhancer blocking) and by segregation of chromatin domains, primarily

insulation of repressive histone mark spread. We investigated whether intra-TAD loops demonstrate these dual insulating properties, canonically ascribed to TADs.

CTCF sites were divided into three groups, based on whether they were predicted to anchor TAD loops or intra-TAD loops, or were not predicted to interact (non-anchor CTCF sites) based on our algorithm. *Figure 3A* shows the extent to which individual CTCF sites within each group show directional interactions towards the loop interior based on Hi-C data, as determined using a chi-squared metric derived from the same directionality index used to predict TADs genome-wide. This inward bias index quantifies the strength of interaction from a 25 kb bin immediately downstream from the CTCF motif, with a positive sign indicating a downstream (inward) bias with regard to motif orientation, that is, towards the center of a TAD/intra-TAD loops in the case of loop anchors. TAD and intra-TAD anchors both show strong directional interactions compared to non-anchor CTCF sites, and TAD anchors show stronger interactions than intra-TAD loop anchors. For the non-anchor CTCF sites, only those sites containing a strong CTCF motif (FIMO score >10) were considered, and the predicted directionality was oriented relative to this motif (as there is no left versus right anchor distinction).

To test specific examples of insulation, we used available high resolution Hi-C for mESC to perform virtual 4C analysis (*Bonev et al., 2017*) at select loop anchors (*Figure 3B*, *Figure 3—figure supplement 1*). This allowed us to visualize the distribution of interactions originating from an intra-TAD loop anchor, and compare them to those originating from an adjacent upstream region (outside of the loop). For virtual 4C viewpoints placed downstream of a *left* intra-TAD loop anchor (i.e., within an intra-TAD loop; IN), interaction reads were shifted in favor of the downstream direction, which comprised 58.8–79.9% of the Hi-C read pairs. This is comparable to the skew observed for 4C-seq experiments performed at TAD anchors (*Guo et al., 2015*; *Gómez-Marín et al., 2015*) (*Figure 3—figure supplement 1A,B*). In contrast, interactions were generally skewed in the upstream direction for viewpoints placed upstream of the same set of intra-TAD loop anchors (*Figure 3—figure supplement 1*, OUT). These shifts in the distribution of interactions further support the insulating nature of these intra-TAD anchors, and were seen both for intra-TADs that are tissue-specific (Albumin, Sox2) and for those that are common (Hnf4a, Scd1) across the three tissues we examined (liver, NPC and mESC cells).

To visualize features of these anchors, aggregate liver Hi-C profiles spanning 1 Mb around each anchor were generated for each group of CTCF sites (*Figure 3C*). Each group was further subdivided into a left (upstream) and a right (downstream) anchor based on its CTCF motif orientation. All non-anchor CTCF sites used for comparison in *Figure 3A and C* were required to contain a CTCF motif to assign directionality, and CTCF peaks were required to be present in a minimum of 2 individual biological replicates. By aggregating many sites, we can visualize the overall interaction properties of each group of sites at high resolution (5 kb bins), revealing features that are much harder to discern at an individual CTCF site (*Figure 3C*). TAD and intra-TAD loop anchors both show strong enrichment of interactions towards the loop interior when compared to CTCF sites that were not predicted to participate in loop formation (non-anchor CTCF sites). Furthermore, intra-TAD anchors show less enrichment of long-range contacts compared to TAD anchors, likely because of their shorter length compared to TADs. This may also explain the lower inward bias scores of intra-TAD loops seen in *Figure 3A*. Depletion of interactions that span across loop anchors (dark blue density above anchor points in *Figure 3C*) was seen for both TAD and intra-TAD anchors, however, this local insulation was substantially greater for TAD anchors. This may be due to the compounding impact of adjacent TAD loops, that is, the end of one TAD is often close to the beginning of an adjacent TAD loop anchor (median distance between TAD anchors = 33.5 kb, *Figure 3—figure supplement 2A*). GO term analysis of genes whose TSS fall within these inter-TAD regions revealed enrichment for housekeeping genes (ribosomal, nucleosome, and mitosis-related gene ontologies), whereas neighboring genes found just within the adjacent TADs were enriched for distinct sets of GO terms (*Figure 3—figure supplement 2B*; *Supplementary file 3B*, *Figure 2—figure supplement 3C*). The nearby but oppositely oriented TAD anchors flanking the inter-TAD regions likely contribute to the more bidirectional interaction pattern for TADs seen in *Figure 3C*. For instance, the left TAD anchor plot in *Figure 3C* shows a well-defined pattern of interaction enrichment downstream from the anchor, but also a more diffuse enrichment upstream contributed by upstream loop anchors located at various distances. In contrast, non-anchor CTCF sites do not show strong directional interactions and only very weak distal contact depletion. CNC-bound regions are predominantly found at

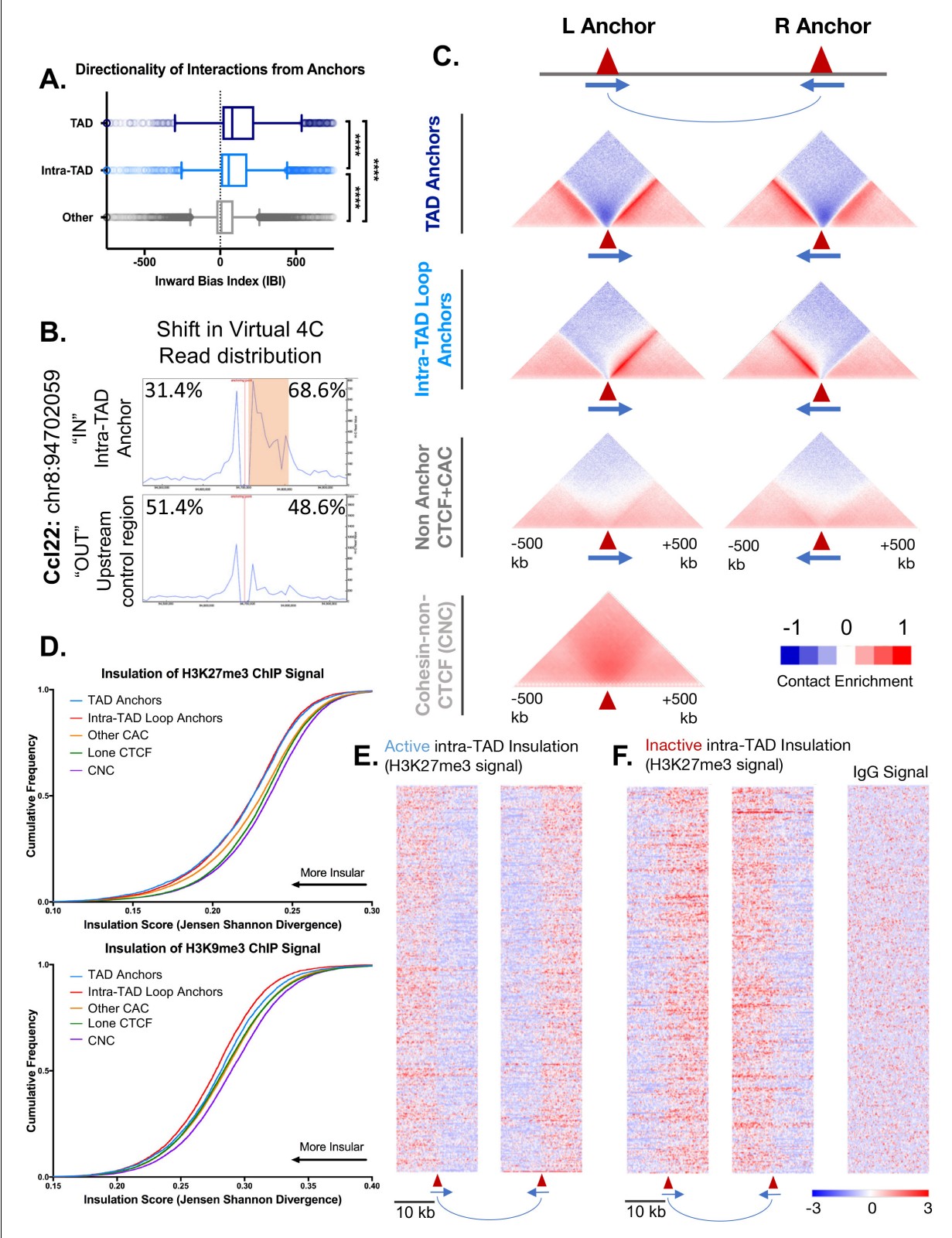

**Figure 3.** Intra-TAD loops show directional interactions and insulate chromatin marks. (**A**) TADs and intra-TAD loops both show a stronger orientation of interactions downstream of the motif than other CTCF-bound regions. TAD anchors also show higher inward bias than intra-TAD loops (p<0.0001, KS t-test for pairwise comparisons). Inward bias is a chi-square-based metric similar to directionality index but defined on a per peak basis and oriented relative to the motif within the anchor/non-anchor peak. For this and all other anchor comparisons, anchors that are shared between TADs and intra-

*Figure 3 continued on next page*

*Figure 3 continued*

TAD loops were excluded from the intra-TAD group to ensure a fair comparison. Anchors shared between TADs and intra-TAD loops were considered as TAD anchors only. (B) Virtual 4C analysis in mESC for a genomic region nearby *Ccl22* on mouse chromosome 8. The data shows a shift in Hi-C read distribution around intra-TAD loop anchors, indicating insulation. mESC Hi-C data from (*Bonev et al., 2017*) was plotted across a 500 kb window surrounding the virtual 4C viewpoint, which is marked by a verticle red line. Viewpoints were selected to be at the start of an intra-TAD loop ('IN') as well as an adjacent upstream control region that does not overlap an intra-TAD loop anchor. The percentages shown indicate the distribution of interaction reads upstream and downstream of the viewpoint, over the 500 kb region, as shown. Orange shading indicates the width of the intra-TAD loop region interrogated. Chromosomal coordinates are for mouse genome mm10. (C) Shown are aggregate plots generated from mouse liver Hi-C data (*Vietri Rudan et al., 2015*) for each set of TAD and intra-TAD loop anchors, for the set of non-anchor CTCF sites listed in *Supplementary file 1C*, and for the set of CNC sites (*Supplementary file 1D*), which serves as a control. In aggregate, TAD and intra-TAD loop anchors show stronger and more directionally-biased interactions (contact enrichment, red) than the non-anchor CTCF bound genomic regions. They also show a greater depletion of distal chromatin interactions (contact depletion, blue). TAD anchors also show greater distal contact enrichment with the anchor and more local contact depletion spanning the anchor than intra-TAD loops. Red triangles indicate locations of left and right loop anchors and blue arrows indicate CTCF motif orientation. Shading indicates an enrichment (red) or depletion (blue) of contact frequency relative to a genome-wide background model. (D) Shown are JSD values for four classes of mutually exclusive CTCF binding sites (TAD anchors, intra-TAD loop anchors, other CAC sites, and CTCF sites lacking cohesin) as well as CNC sites, which are primarily found at enhancers. TAD and intra-TAD loop anchors show greater insulation of the repressive histone marks as measured by Jensen Shannon divergence between H3K27me3 and H3K9me3 ChIP-seq signal upstream and downstream of the anchor region. (E) Shown are the top 500 active insulated intra-TAD loops, based on high H3K27me3 ChIP-seq signal outside the intra-TAD loop (red), and low H3K27me3 signal within the intra-TAD loop (blue). Data are expressed as a Z-score of the H3K27me3 signal per bin relative to all H3K27me3 signals within a 20 kb widow centered on all CTCF-bound regions. (F) Shown are the top 500 inactive insulated intra-TAD loops, based on high signal H3K27me3 signal inside the intra-TAD loops (red) and low H3K27me3 signal in neighboring regions (blue). Signal is shown as a Z-score of H3K27me3 signal, as in E. At right is shown the IgG signal distribution as a negative control for the upstream anchors of inactive intra-TAD loops (see *Figure 3—figure supplement 2D* for all IgG signal panels).

DOI: https://doi.org/10.7554/eLife.34077.014

The following figure supplements are available for figure 3:

**Figure supplement 1.** Direct evidence of insulation from the asymmetric read distributions for virtual 4C viewpoints anchored at intra-TAD loop anchors or adjacent upstream regions.
DOI: https://doi.org/10.7554/eLife.34077.015

**Figure supplement 2.** Additional features of intra-TADs and their insulation.
DOI: https://doi.org/10.7554/eLife.34077.016

enhancers (*Faure et al., 2012*) and do not show any discernable patterns of insulation or focal and directional interactions (*Figure 3C*, bottom). Thus, the weak contact depletion spanning each of the CTCF-containing groups shown in *Figure 3C* is likely real, and not an artifact of the background model or other noise.

To determine if intra-TAD loops share with TADs the ability to block histone mark spread and establish broad, insulated chromatin domains of activity and repression, we analyzed the distribution of two repressive marks in relation to CTCF binding sites: H3K27me3 and H3K9me3. An insulation score based on Jansen Shannon Divergence (JSD) (*Fuglede and Topsoe, 2004*) was calculated for ChIP signal distribution within a 20 kb window around each CTCF peak. A low JSD value indicates less divergence from a string representing perfect insulation (i.e., high signal on one side of peak, and low or no signal on the other). This scoring allows for a direct comparison of different classes of CTCF sites, or other TF-bound sites, in terms of their insulation properties. In addition to TAD and intra-TAD loop anchors, we examined three other sets of sites as controls: other CAC sites, sites bound by CTCF alone, and CNC sites. *Figure 3D* shows the cumulative distribution of JSD scores for each set of sites, where a leftward shift indicates greater insulation across the site for the chromatin mark examined. For H3K27me3 signal distribution, TAD and intra-TAD loop anchors showed the greatest insulation, but were not significantly different from each other (Kolmogorov-Smirnov (KS) test; $p=0.52$). The same general trend was observed for H3K9me3 signal insulation; however, intra-TAD loop anchors actually showed greater insulation than TAD anchors and all other groups (KS test; $p<0.001$). CNC sites consistently showed the least insulation of both repressive histone marks, as expected. Sites where CTCF is bound alone showed a small but significant increase in insulation compared to CNC sites, as did the CAC group. As a control, the distribution of IgG signal (input signal) showed much less insulation overall and no significant differences between the various classes of CTCF/cohesin binding sites (*Figure 3—figure supplement 2C*).

*Figure 3E and F* show heat map representations of H3K27me3 ChIP-seq signal around the top 500 active and top 500 inactive intra-TAD loops, based on a ranked list of JSD insulation scores. These represent intra-TAD loops that have significantly lower (or higher) H3K27me3 signal in the loop interior based on the combined rank of JSD insulation scores for each anchor, respectively ($p<0.05$, two-sided t-test). For example, *Figure 3F* shows intra-TAD loops with lower H3K27me3 signal within the loop than in neighboring regions (and thus the left anchor transitions from high to low signal, while the right anchor transitions from low to high signal). No such pattern was seen for the IgG (control) ChIP-seq signals for these same regions, indicating this is not an artifact of the sequence read mappability of these regions (*Figure 3F.*, *Figure 3—figure supplement 2D*).

## Impact of intra-TAD loops on *cis* regulatory elements in mouse liver

### Classifying open chromatin regions and defining super-enhancers in mouse liver

Many TADs, and also intra-TAD loops, are structurally conserved across tissues, yet the activity of enhancers and promoters contained within these looped structures is highly tissue-specific (*Dowen et al., 2014*; *Heidari et al., 2014*). Accordingly, it is important to understand the ability of TADs and intra-TAD loops to insulate active enhancer interactions, that is enhancer-blocking activity. To address this issue, we examined the distribution of CTCF and cohesin binding sites in relationship to promoters and enhancers across the genome, as well as the impact of TADs and intra-TAD loops on their associated genes and enhancers.

We previously identified ~70,000 mouse liver DNase hypersensitive sites (DHS), whose chromatin accessibility is in part determined by TF binding and their flanking histone marks (*Sugathan and Waxman, 2013*; *Ling et al., 2010*). To assign a function for each DHS, we classified each DHS according to the ratio of two chromatin marks, H3K4me1 and H3K4me3, which are respectively associated with enhancers and promoters (*Wang et al., 2008*). The ~70,000 DHS were grouped into five classes: promoter, weak promoter, enhancer, weak enhancer and insulator (*Figure 4A*, *Figure 4—figure supplement 1A*; see Materials and methods). Promoter-DHS were defined as DHS with a H3K4me3/H3K4me1 ratio $\geq$1.5, and enhancer-DHS by a H3K4me3/H3K4me1 ratio $\leq$0.67 (as first described in [*Hay et al., 2016*]). DHS with similar signals for each mark (H3K4me3/H3K4me1 ratio between 0.67 and 1.5) were designated weak promoter-DHS, based on their proximity to TSS and comparatively lower expression of neighboring genes (*Figure 4—figure supplement 1B,C*). DHS with low signal for both marks were classified as weak enhancers, or as insulators, in those cases where they overlapped a CTCF peak with a comparatively strong ChIP-seq signal (*Figure 4—figure supplement 1A*). Using this simplified five DHS class model, we observed that enhancer-DHS bind cohesin largely in the absence of CTCF, while promoter-DHS are bound by both CTCF and cohesin (*Figure 4B*). Additionally, H3K27ac is enriched at promoter-DHS and enhancer-DHS but not at weak enhancer-DHS, which are less open (lower DNase-seq signal) (*Figure 4B*) and more distal (*Figure 4—figure supplement 1B*). In contrast to enhancer-DHS, insulator-DHS have a well-defined bimodal distribution of tissue-specific vs. tissue-ubiquitous sites based on comparisons across 20 ENCODE tissues (*Figure 4—figure supplement 1D*). This supports the proposal that insulators are a unique class of intergenic regions, and not simply enhancers bound by CTCF.

A subset of intra-TAD CAC loops are well characterized as insulators of tissue-specific genes with highly active enhancer clusters, termed super-enhancers (*Dowen et al., 2014*; *Whyte et al., 2013*). To determine if some intra-TAD loops correspond to these 'super-enhancer domains', we first identified super-enhancers in mouse liver. We used 19 publicly available mouse liver H3K27ac ChIP-seq datasets to score clusters of individual enhancer-DHS + weak enhancer-DHS identified above (*Figure 4C*). Super-enhancers were identified separately in male and female liver, as well as in male liver at various circadian time points, to take into account these three key sources of natural variation in gene expression in mouse liver (*Sugathan and Waxman, 2013*; *Fang et al., 2014*). In total, we identified 503 core super-enhancers, that is, super-enhancers that show strong signal regardless of sex or time of day (*Figure 4—figure supplement 2A*). Core super-enhancers represent 14.1% of all enhancer regions in the genome (6680 of 47,372 constituent enhancers + weak enhancers), and 2.8% of all enhancer clusters, defined as groups of enhancers within 12.5 kb of one another (503 of all 17,964 enhancer clusters).

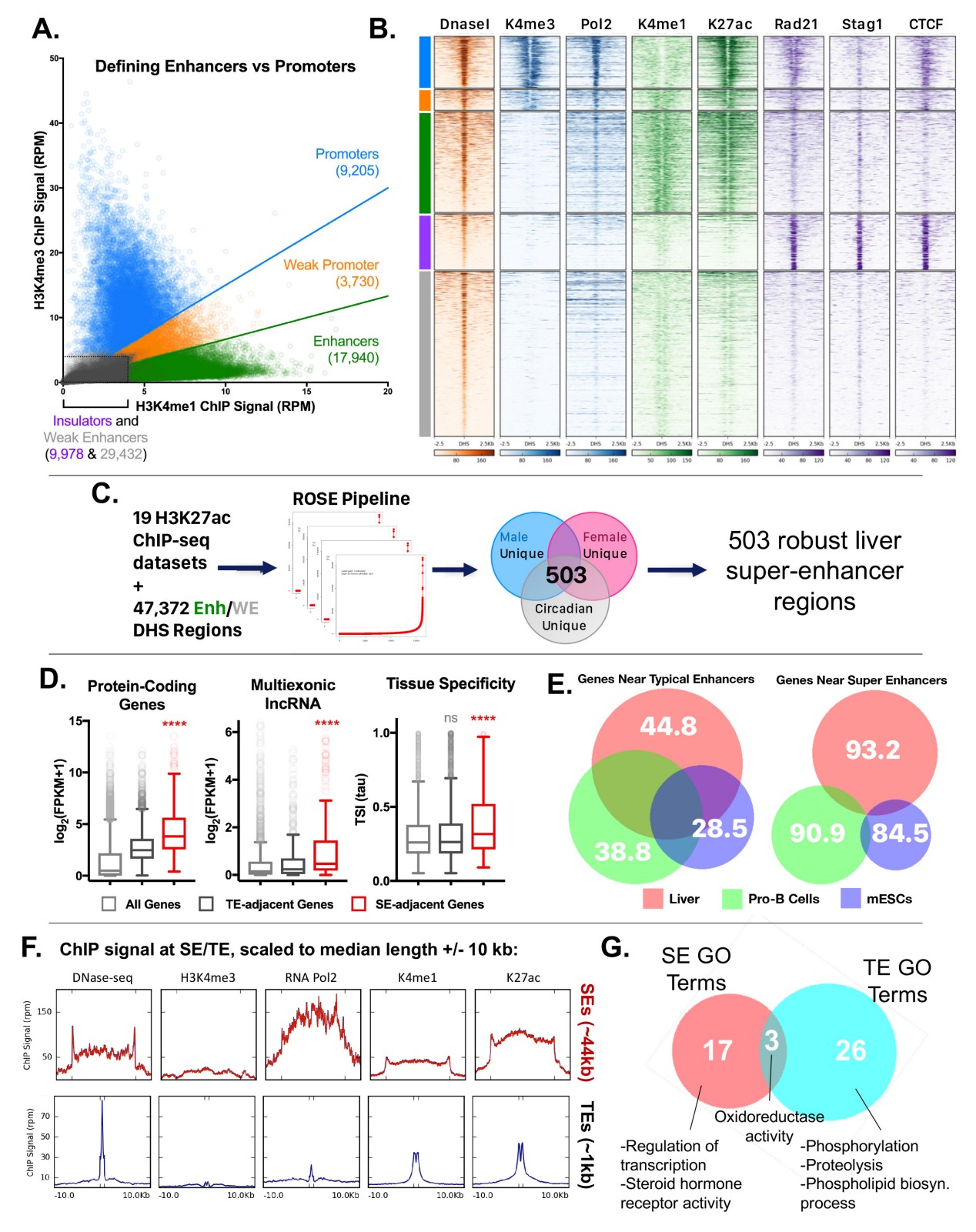

**Figure 4.** Categorization of DHS-based regulatory elements in mouse liver. (**A**) Classification of set of ~70,000 open chromatin regions (DHS) identified in adult male mouse liver, based on relative intensities for a combination of H3K4me1 and H3K4me3 marks, and CTCF ChIP-seq data. Based on the combinatorial signal from these three datasets, five groups of DHS were identified: promoter-DHS, weak promoter-DHS, enhancer-DHS, weak enhancer-DHS, and insulator-DHS, as described in Materials and methods and in *Figure 4—figure supplement 1A*. (**B**) Shown is a heatmap

*Figure 4 continued on next page*

*Figure 4 continued*

representation of the simplified five-class DHS model shown in panel A, which captures features such as CNC enrichment at enhancers and K27ac enrichment at enhancers and promoters, with additional features described in *Figure 4—figure supplement 1A*. Color bar at the *left* matches colors used in panel A. (C) Scheme for using 19 published mouse liver H3K27ac ChIP-seq datasets to identify a core set of 503 liver super-enhancers using the ROSE software package (*Supplementary file 2B*). These 503 super-enhancers were identified in all 19 samples, indicating they are active in both male and female liver, and across multiple circadian time points. Enh, enhancer, WE, weak enhancer. (D) Genes associated with super-enhancers (SE) are more highly expressed (log2(FPKM +1) values) than genes associated with typical enhancers (TE), for both protein coding genes and liver-expressed multi-exonic lncRNA genes. The super-enhancer-adjacent genes are also more tissue specific (higher Tau score) than typical enhancer-adjacent genes. ****, KS t-test, p<0.0001 for pairwise comparisons of SE-adjacent genes vs. TE-adjacent genes. (E) Venn diagrams show substantial overlap between typical enhancer gene targets across tissues (liver, ESCs, ProB cells), but limited overlap between super-enhancer adjacent genes (within 10 kb of the super-enhancer) for the same tissues. The numbers represent the percent of genes targeted in a given tissue by the indicated class of enhancer (typical enhancers or super-enhancers) that are not targets of the corresponding class of enhancers in the other two tissues. For example, 93.2% of genes targeted by liver super-enhancers are not targeted by the set of super-enhancers identified in either Pro-B or mouse ESCs. Gene targets of each enhancer class were identified by GREAT using default parameters, then filtered to keep only those ≤10 kb from the enhancer. (F) ChIP and DNase-seq signal at typical enhancers and super-enhancers, scaled to their median length (1 kb and 44 kb respectively; indicated by distance between hash marks along the x-axis) flanked by 10 kb up- and down-stream. Super-enhancers show much greater accumulation of RNA polymerase 2, despite little or no apparent enrichment for the promoter mark H3K4me3. (G) Super-enhancers (SE) target distinct categories of genes than typical enhancers (TE) in mouse liver. Thus, while GO terms such as oxidoreductase activity are enriched in the set of gene targets for both classes of enhancers, only super-enhancers are enriched for transcription-regulated terms (e.g., Regulation of transcription, Steroid hormone receptor activity) (*Supplementary file 2C, D*). Numbers represent the overlap of GO terms (either Molecular Function or Biological Process) in any DAVID annotation cluster (with an enrichment score >1.3) enriched for genes regulated by either typical enhancers or super-enhancers.

DOI: https://doi.org/10.7554/eLife.34077.017

The following figure supplements are available for figure 4:

**Figure supplement 1.** Characteristics of five classes of DHS in mouse liver.
DOI: https://doi.org/10.7554/eLife.34077.018

**Figure supplement 2.** Features of super-enhancers and single-TSS intra-TAD loops.
DOI: https://doi.org/10.7554/eLife.34077.019

---

Both protein coding and lncRNA genes that neighbor super-enhancers are more highly expressed and tissue-specific when compared to all genes, or to all genes neighboring typical enhancers (KS test p-value<0.0001; *Figure 4D*). Consistent with this tissue specificity, only 6.8% of genes proximal to liver super-enhancers are targets of super-enhancers in mESCs or pro-B cells (*Figure 4E*), whereas 55.2% of genes proximal to liver typical enhancers are proximal to typical enhancers in the other two cell types. Super-enhancers showed much greater accumulation of RNA polymerase 2, despite the lack of the promoter mark H3K4me3 (*Figure 4F*) and are transcribed to yield eRNAs (*Figure 4—figure supplement 2B*). GO terms associated with genes targeted by either typical enhancers or super-enhancers are enriched for liver functions (such as oxidoreductase activity), however, super-enhancer target genes also show enrichment for transcription regulator activity and steroid hormone receptor activity (*Figure 4G*). These data support the model that super-enhancers drive high expression of select liver-specific genes, including transcriptional regulator genes (*Supplementary file 2C,D*).

Strikingly, 72.2% of core super-enhancers (363/503) overlap either an intra-TAD loop or a TAD that contains only a single active gene (defined as a gene expressed at FPKM ≥1 and with a promoter-DHS within 5 kb of the TSS; *Supplementary file 1B* and *Supplementary file 2B*) (see, e.g., *Figure 5C and D*, below). By comparison, only 43.6% (17,742/40,692) of typical enhancers are insulated in a similar manner (data not shown). We also observed an enrichment of single-TSS intra-TAD loops (n = 3,142) over a random shuffled set (*Figure 4—figure supplement 2C*), which could represent tissue-specific genes that are regulated by super enhancers in liver or in other tissues. Genes within these single-TSS intra-TAD loops (*Supplementary file 1B*) were enriched for ontologies related to transcriptional regulation and phosphorylation (*Figure 4—figure supplement 2D*, *Supplementary file 3E*). This is consistent with a model of intra-TAD loops as functionally inducible units of gene expression, allowing selective transcription in a given tissue or in response to cell signaling events.

To determine the impact of intra-TAD loops on the expression of genes with neighboring super-enhancers, we considered two possible gene targets for each super-enhancer, with the requirement that the TSS of each gene target be within 25 kb of one of the individual enhancers that constitute the overall super-enhancer: one gene target is located within the intra-TAD loop, and the other

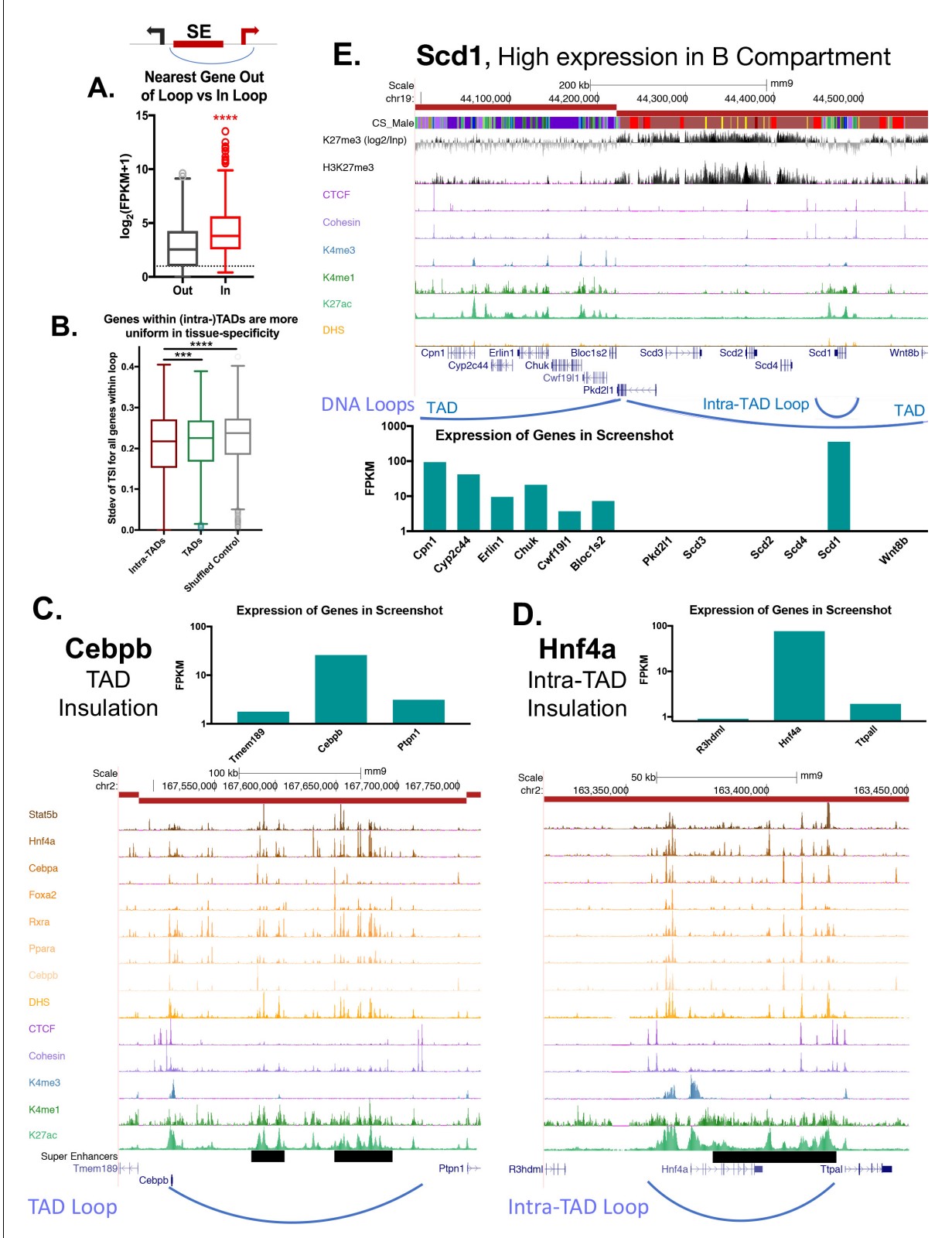

**Figure 5.** Impact of intra-TAD loops on gene expression. (**A**) Two possible gene targets were assigned for each super-enhancer within an intra-TAD loop, one target gene for which the TSS is within the intra-TAD loop and another target gene for which the TSS is outside of the intra-TAD loop but is within 25 kb of the intra-TAD loop anchor. Box plots show that gene targets within an intra-TAD loop are significantly more highly expressed than the alternative, linearly more proximal, gene target. (**B**) Shown is the standard deviation in Tau values (tissue-specificity index) of genes whose TSS's are

*Figure 5 continued on next page*

*Figure 5 continued*

within TADs or intra-TAD loops that contain at least three TSS. Genes within intra-TAD loops tend to be more uniformly tissue-specific or tissue-ubiquitous when compared to all genes within TADs, or when compared to a shuffled set of random regions matched in size to intra-TADs. Thus, sets of three or more genes within intra-TAD loops are consistently either more or less tissue specific than random clusters of genes within the same sized genomic spans. (C–D) TAD and intra-TAD loops insulate a subset of super-enhancers (black horizontal bars) with key liver genes, allowing high expression of genes such as the TFs *Cebpb* and *Hnf4a*, relative to their immediate neighbors. *Cebpb* is an example at the TAD scale, while *Hnf4a* shows an intra-TAD loop. In both cases, the most linearly proximal gene is outside the TAD or intra-TAD loop and is expressed at a lower level than the loop-internal genes (and presumptive gene target). (E) Shown is a UCSC genome browser screenshot of a transition from an active to a repressed TAD, with the expression of genes within the region shown in a bar graph, below. Insulated intra-TAD loops allow for expression of select gene targets within otherwise repressed genomic compartments. The obesity-related gene *Scd1* is insulated in an intra-TAD loop and is the only liver expressed gene in its TAD (FPKM >100). H3K27me3 marks are shown both as reads per million signal track (below) and as signal over an IgG input control (above), expressed as $\log_2[(H3K27me3\ signal) / (Input\ signal)]$.

DOI: https://doi.org/10.7554/eLife.34077.020

gene target crosses an intra-TAD loop anchor (*Figure 5A*, scheme at *top*; genes inside (red) and genes outside (black) of the intra-TAD loop). We hypothesized that the true gene target of the super-enhancer will be more highly expressed. Indeed, we found that genes neighboring super-enhancers and found within the same intra-TAD loop are significantly more highly expressed than the alternative potential gene targets, located outside of the intra-TAD loop (*Figure 5A*). Similarly, when comparing the tissue specificity of genes within TADs and intra-TAD loops to a random shuffled set of regions, we observed less variance in the Tau value (index of tissue specificity) for genes within TAD or intra-TAD loops compared to the shuffled set (*Figure 5B*; KS test p-value<0.0001). Thus, groups of genes within intra-TAD loops are more uniformly tissue specific (as in the case of some super-enhancer-adjacent genes) or tissue ubiquitous (as in the case of groups of housekeeping genes). Examples are shown in *Figure 5C and D*, which illustrate the impact of a TAD loop on the expression of *Cebpb* and the impact of an intra-TAD loop on the expression of *Hnf4a*. The TSS of two other nearby genes, *Ptpn1* and *Ttpa1,* are closer in linear distance to the adjacent super-enhancer than *Cebpb* and *Hnf4a*, respectively, however, any super-enhancer-promoter interactions involving *Ptpn1* and *Ttpa1* would need to cross a TAD or intra-TAD loop boundary. In both cases, the genes within the super-enhancer-containing TAD or intra-TAD loop are expressed at least 10-fold higher than genes outside the loop. Therefore, based on the 3D-structure imposed by these TADs and intra-TAD loops, one predicts that the super-enhancers are restricted from interacting with *Ptpn1* or *Ttpa1*, in agreement with the comparatively low expression levels of those genes.

Given the ability of intra-TAD loops to insulate repressive histone marks (*Figure 3D–3F*), we considered whether intra-TAD loops enable high expression of genes within otherwise repressed genomic compartments. As seen in *Figure 1H*, a minority of genes within inactive TADs are expressed (939 genes expressed at FPKM >1). The obesity-related gene stearoyl-CoA desaturase-1 (*Scd1*) is one such gene. *Figure 5E* shows a transitional TAD boundary, with genes in the upstream TAD expressed and associated with low levels of H3K27me3 repressive histone marks. Six genes are in the downstream TAD, but only one of these genes, *Scd1*, is expressed (*Figure 5E*, *bottom*). The high expression of *Scd1* (FPKM >100) can be explained by its localization in an active intra-TAD loop that is insulated from the repressive mark H3K27me3 compared to the rest of the TAD. This same structural organization was seen for 291 of the 939 expressed genes found in inactive TADs (*Figure 1H*, above), which are contained within intra-TAD loops. It is unclear what other mechanisms allow for selective expression of the other 648 genes (*Supplementary file 1E*).

## 4C-seq analysis of super-enhancer contacts at Alb promoter

To test directly for the insulation of an intra-TAD loop containing a super-enhancer, we performed 4C-seq analysis for the promoter of albumin (*Alb*), the most highly expressed gene in adult mouse liver. 4C-seq is designed to identify all chromatin contacts originating from a single genomic region (the promoter of *Alb* in this case), known as the 4C viewpoint. Using 4C-seq, we captured many highly specific, reproducible interactions with the *Alb* promoter, a majority of which are localized across an upstream ~50 kb region (*Figure 6A*, *Figure 6—figure supplement 1A,B*). 40% of the interactions are localized within DHS that are constituent enhancers of the *Alb* region super-enhancer. Furthermore, >80% of chromatin contacts are within the *Alb* super-enhancer, and >98%

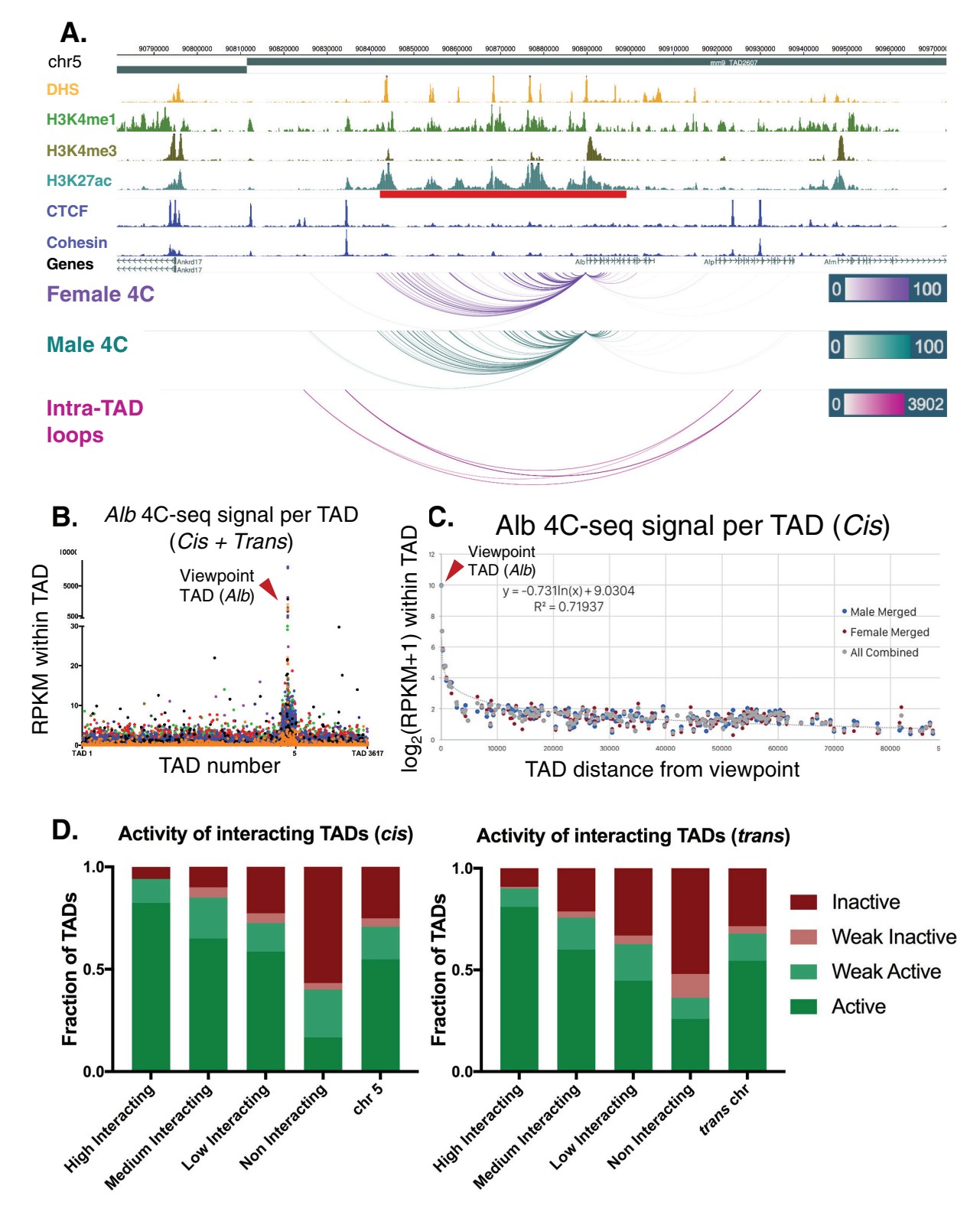

**Figure 6.** *Alb* 4C-seq exemplifies intra-TAD insulation and super-enhancer interaction. (**A**) The *Alb* promoter makes multiple directional contacts with the adjacent super-enhancer region in both male and female mouse liver, as determined by 4C-seq with a viewpoint at the *Alb* promoter. All reproducible interactions occur within the TAD loop containing the *Alb* TSS and its super-enhancer (red bar beneath H3K27ac track), and all but two contacts in male liver occur within the predicted intra-TAD loops (pink). 4C-seq interaction scores are shown as –log10(pval) values across replicates, as

*Figure 6 continued on next page*

*Figure 6 continued*

calculated by R3C-seq (see Materials and methods). Also see *Figure 6—figure supplement 1*. (B) The 4C-seq interaction signal within the *Alb* TAD is orders of magnitude above the background signal and generally decays with distance. Far-*cis* and *trans* interactions are represented on a per TAD basis, expressed as RPKM per TAD, to control for sequencing depth and TAD length. The overall background within mouse chromosome five is significantly higher than all *trans* chromosomes; immediately adjacent TADs also show higher 4C-seq signal than the overall cis background. The 4C-seq signal decayed to background levels within ~3 TADs of the *Alb* viewpoint TAD. Each data point represents a single TAD and each color represents a 4C-seq replicate. (C) Background model used for distal *cis* interactions, showing a rapid decay in per TAD signal intensity. Each data point represents a single TAD along chromosome 5. (D) Distal *cis* and *trans* TADs that highly interact with the *Alb* promoter tend to be active TADs, while a majority of the TADs that interact less than the background model are predicted to be inactive. A simple inverse logarithmic decay of signal per TAD was used to determine the background signal along the *cis* chromosome, while the 4Cker package was used to determine high, medium, low, and non-interacting TADs in trans based on a hidden markov model with adaptive windows better suited for low signal regions.

DOI: https://doi.org/10.7554/eLife.34077.021

The following figure supplement is available for figure 6:

**Figure supplement 1.** *Alb* 4C-seq replicates and cis/trans 4C-seq signal distribution.

DOI: https://doi.org/10.7554/eLife.34077.022

are contained within the *Alb* intra-TAD loop, in both male and female liver. These interactions become more diffuse with increasing genomic distance from the viewpoint, which may represent dynamic interactions, or alternatively, may reflect averaging across a heterogeneous cell population. Comparing the combined interaction profiles between male and female livers, we observed highly reproducible results, with 92.9% of male interactions also present in female liver (*Figure 6—figure supplement 1C*).

Looking beyond local interactions, we observed 4C interaction frequencies that span several orders of magnitude, going from local (intra-TAD) to *cis* (intra-chromosomal) and *trans* (genome-wide) interactions. Thus, for the *Alb* promoter, the 4C signal per TAD was >1000 RPKM for local interactions, >100 RPKM for *cis* interactions within 3 TADs adjacent to the viewpoint, and ~10 RPKM beyond that (*Figure 6B*). *Trans* interactions were almost exclusively <10 RPKM (*Figure 6B*). Using a separate background model for far-*cis* and *trans* 4C signals, we categorized TADs as either high, medium, low, or non-interacting (see Materials and methods). As *Alb* is the most highly expressed gene in liver and is proximal to a strong super-enhancer, we expected that distal interacting regions would also be active genomic regions, as proposed in the transcription factory model of nuclear compartmentalization (*Iborra et al., 1996*). Indeed, we found that >80% of the distal high interact-ing TADs (both far-*cis* and *trans*) were active TADs, and >90% were either active or weakly active TADs. In contrast, only 16.6% of the non-interacting TADs in cis and 25.9% of those in trans were active TADs (*Figure 6B,D*). Furthermore, genes in the interacting regions are more highly expressed than genes in *Alb* 4C non-interacting regions, and the vast majority are found in active TADs (*Figure 1F*), as determined by analysis of the Hi-C data alone (*Figure 6—figure supplement 1D,E*).

## Discussion

We present, and then validate in multiple mouse and human cell models, a computational method that uses 2D (ChIP-seq) and 1D (DNA sequence) information to predict 3D-looped intra-TAD struc-tures anchored by cohesin and CTCF (CAC sites), and we provide evidence that the intra-TAD loops predicted underpin a general mechanism to constrain the interactions between distal enhancers and specific gene targets. While select instances of CAC-mediated loop insulation within TADs have been described (*Dowen et al., 2014*; *Willi et al., 2017*; *Hanssen et al., 2017*), our work establishes that this phenomenon is a more general feature of genomic organization and regulation than previ-ously appreciated. The intra-TADs described here are nested, CAC-anchored loops whose formation may be a result of extrusion complex pausing within larger domains (i.e., TADs); these loops act to constrain the promoter contacts available to a given distal enhancer, and correspondingly, the distal enhancer contacts available to a given promoter (*Hnisz et al., 2016*). We also provide evidence that the loop-forming CTCF sites, but not other CTCF sites, are highly insular. This insulation is apparent from the blockage of repressive histone mark spread and by the inhibition of chromatin contacts across intra-TAD loop and TAD boundaries. The impact of this insulation is highlighted for super-enhancer regions, such as the super-enhancer upstream of *Alb*, where local insulation by CAC-

anchored intra-TAD loops both enables and constrains strong near-*cis* interactions, which facilitate the high expression of *Alb* and presumably also other liver-expressed genes regulated by super-enhancers. Weaker *trans* interactions with distal active regions were also observed, and are likely driven by a distinct mechanism, such as aggregation of transcription factories or super-enhancers (*Rao et al., 2017*; *Osborne et al., 2004*).

Genomic interactions occur at three levels: (1) compartmentalization, where inactive regions localize to the nuclear periphery and active chromatin compartments aggregate toward the center of the nucleus in *cis* or *trans* in a largely cohesin-independent manner, as proposed in the transcription factory model (*Rao et al., 2014*; *Seitan et al., 2013*; *Osborne et al., 2004*; *Lieberman-Aiden et al., 2009*); (2) CAC-dependent looping, which generates many tissue-invariant scaffolds along the linear chromosome (*Dixon et al., 2012*; *Sanborn et al., 2015*; *Hnisz et al., 2016*); and (3) enhancer-promoter looping within CAC-loops, which may be directed by cohesin non-CTCF (CNC) sites, mediator, or tissue-specific TFs (*Dowen et al., 2014*; *Kagey et al., 2010*; *Faure et al., 2012*). If TADs define the broad domain within which a cohesin-driven extrusion complex generally operates, then we have presented a simple method to identify loops within this region that form as a result of dynamic loop extrusion movement and pausing at additional loop anchors. We have used the term intra-TAD loops, also referred to as sub-TADs, to highlight their subdivision of TAD-internal genomic space, although they are functionally similar to loop domains, isolated cliques, and insulated neighborhoods, which tend to overlap or be contained within TADs (*Sanborn et al., 2015*; *Rao et al., 2014*; *Hnisz et al., 2016*). Our computational method cannot predict CTCF-independent loops, such as those mediated cohesin alone (enhancer-promoter loops), although such loops are likely constrained by CAC driven intra-TADs, as was highlighted by our *Albumin* 4C-seq results.

The method for CAC-mediated intra-TAD loop identification described here builds on the strong preference for inward-facing CTCF motifs evident from high resolution Hi-C data (*Sanborn et al., 2015*; *Rao et al., 2014*), and will be most useful for the identification of intra-TAD CAC loops for the large number of cell lines and tissues that lack high resolution Hi-C data. In these cases, intra-TAD loop domains cannot be identified because there is not sufficient local enrichment to calculate a corner score with the arrowhead algorithm (*Rao et al., 2014*). Further, while we used TAD boundaries from standard resolution liver Hi-C data to filter out longer CAC loops, the frequent conservation of TADs across both tissues and species (*Dixon et al., 2012*; *Vietri Rudan et al., 2015*) broadens the applicability of our method to cell types, and perhaps to new species, for which Hi-C data is not available and TAD boundaries have not been determined. Thus, even in the absence of TAD coordinates, our method identifies TAD and intra-TAD looping events, which may provide an invaluable first approximation for understudied organisms. As we have tuned our parameters to identify loop structures comparable in size and number to those found previously in mouse and human models, the parameters used to filter an initial set of loop anchors may need to be adjusted for other model organisms.

We have used both CTCF and cohesin peak strength as the primary predictor of intra-TAD loop strength, which is a reasonably good predictor of interactions (*Sanborn et al., 2015*; *Oti et al., 2016*). An alternative machine learning approach to predicting CTCF/cohesin-mediated interactions, posted as an on-line preprint during review of our manuscript (*Kai et al., 2017*), uses data from up to 77 genomic-derived features to predict CTCF-mediated loops in three human cell lines. A key finding from this work was that cohesin strength was consistently the most predictive feature of CTCF loops, followed by CTCF binding strength (*Kai et al., 2017*). This method also captures loops that lack convergent CTCF motif orientation, which represent as few as 8% of the total for loop domains (*Rao et al., 2014*), or as many as 20% in the case of Insulated Neighborhoods (*Ji et al., 2016*). However, the identification of this subset of loops comes at the expense of requiring a minimum of 16 features for a given cell type, whereas our approach only requires three features (CTCF motif, CTCF ChIP-seq, and cohesin ChIP-seq data). Importantly, the three features used by our method represent 3 of the top four predictive features identified in (*Kai et al., 2017*).

The computational method presented here, which was validated in both mouse and human cell models, provides a practical alternative to using high resolution Hi-C libraries for the identification of intra-TAD loops. High resolution Hi-C requires extremely deep sequencing, which is costly, both in terms of computational and experimental laboratory resources, and has only been achieved for a small number of cell lines (*Rao et al., 2014*; *Jin et al., 2013*; *Bonev et al., 2017*). Strategies to reduce the need for extreme deep sequencing to identify interactions at high resolution have been

proposed (*Weinreb and Raphael, 2016*; *Martin et al., 2015*; *Zhang et al., 2017*), and are beginning to make higher resolutions possible in more systems, however, the sequencing depth and cost will likely remain out of reach for many labs. Antibody enrichment for select genomic regions followed by chromosome conformation capture, as implemented in ChIA-PET, is an experimental alternative to intra-TAD prediction. ChIA-PET and other 3C-based antibody enrichment methods select for genomic regions that are highly bound by the protein(s) of interest (e.g., CTCF and cohesin), and can therefore identify 'many to many' interactions, instead of the 'all to all' interactions identified by Hi-C; these methods are therefore more practical than Hi-C, in terms of their sequencing depth requirements (*Fullwood et al., 2009*). However, ChIA-PET still requires ~10 fold more extensive deep sequencing per sample (~400 million reads) than is needed to obtain the CTCF and cohesin ChIP-seq data utilized in our computational analysis to identify intra-TAD loops. Further, as ChIA-PET uses antibody to select for genomic regions bound by CTCF and/or cohesin, it is difficult to differentiate strength of antibody binding to the anchor proteins from strength of chromatin interaction between the anchors. Of the various CTCF loops described in the literature, insulated neighborhoods are most similar to the intra-TAD loops described here. Insulated neighborhoods are proposed to rectify the observation of smaller and more abundant loops, evident in ChIA-PET datasets, with the established TAD model of large loops from Hi-C experiments (*Dowen et al., 2014*; *Rao et al., 2014*; *Tang et al., 2015*).

The TAD and intra-TAD loop anchors identified here together comprise 27% of all liver CTCF binding sites, consistent with the 30% of murine ESC CTCF peaks that overlap insulated neighborhood anchor regions (*Hnisz et al., 2016*). The precise mechanism that differentiates these CTCF sites, which anchor intra-TAD and TAD loops, from the larger number of non-anchor CTCF binding sites present in any given tissue is unknown. Further, it is unclear what role the typically weaker remaining ~70% of CTCF sites play in organizing the nucleus. Some of these non-(intra-)TAD anchor CTCF sites may serve other, unrelated functions, given the ability of CTCF to interact with other TFs, bind RNA, and regulate splicing mechanics (*Lutz et al., 2000*; *Ross-Innes et al., 2011*; *Sun et al., 2013*; *Saldaña-Meyer et al., 2014*; *Shukla et al., 2011*). Alternatively, some of these CTCF sites may anchor loops present in only a minority of cells in the population analyzed, which would account for their overall weaker signals. Early single cell Hi-C experiments suggested that TADs are present in virtually all individual cells (*Nagano et al., 2013*), however, more recent studies indicate cell-to-cell variability in TADs within a given cell population, although the presence of distinct active and inactive genomic compartments is common across most individual cells (*Stevens et al., 2017*; *Wang et al., 2016*). Truly high-resolution elucidation of single cell intra-TAD structures may not be possible due to the intrinsic limitation of two potential ligation events per fragment in any given cell.

We found that CAC sites are found at insulators and also at promoters, which we defined as DNase hypersensitive sites (DHS) with high a histone-H3 K4me3/K4me1 ratio, whereas CNC sites are primarily at enhancers and weak enhancers. Others find that promoters, when defined as the set of all TSS upstream regions (including those not at a DHS), are bound by cohesin alone (*Kagey et al., 2010*; *Faure et al., 2012*). Further, we found that CTCF-bound open chromatin regions distal to promoters (insulator-DHS) show features that distinguish them from other classes of open chromatin (promoter-DHS and enhancer-DHS), including the absence of enhancer marks and their general conservation across tissues. Thus, these insulator-DHS regions are not simply enhancers with CTCF bound. Supporting this, insulator regions consistently show less intrinsic enhancer activity than weak enhancers in *in vivo* enhancer screens (*Vanhille et al., 2015*). It is less clear what role CTCF binding in the absence of cohesin plays in the nucleus, as we found such sites lack insulating activity and also lack strong directional interactions. As CTCF binding is always intrinsically directional, due to its non-palindromic motif, the absence of directional interactions from CTCF-non-cohesin sites suggests that the directionality of interactions with CTCF sites at TAD and intra-TAD loop anchors is conferred by other factors associated with the extrusion complex, such as cohesin (*Sanborn et al., 2015*; *Fudenberg et al., 2016*) or Top2b (*Uusküla-Reimand et al., 2016*). However, our findings suggest that the interactions of Top2b involve binding to cohesin, and not CTCF, as indicated by the high frequency of CNC sites bound by Top2b vs. very low frequency of Top2b binding at CTCF-non-cohesin sites (*Figure 2—figure supplement 3B*). Furthermore, binding by Top2b does not distinguish TAD anchors from intra-TAD loop anchors. Indeed, by all metrics tested, we found no TF or motif that differentiates TAD anchors from intra-TAD loop anchors, although the existence of some unknown differentiating factor cannot be ruled out. Cohesin can stabilize large protein complexes

comprised of up to 10 distinct TFs at enhancers (*Faure et al., 2012*), and could thus facilitate the binding of other unknown proteins to the loop extrusion complex.

Cohesin is continuously recycled throughout the genome by loading and release factors (*Busslinger et al., 2017*), and so it is unclear how insulator activity is effectively maintained at TAD and intra-TAD loop anchors in such a dynamic environment. We found that CNC sites, which are primarily found at enhancers, consistently show the least insulation of repressive histone marks, just as they show the least insulation of chromatin contacts. This provides further evidence that TAD and intra-TAD loop anchors are functionally unique sites, and are not a moonlighting feature of CTCF bound to enhancer regions. Furthermore, while enhancers are strongly enriched for genetic non-coding variants, genetic variations at loop anchors are rare (*Hnisz et al., 2016*). Mutations that occur at loop anchors can result in dramatic phenotypes like polydactyly or tumorigenesis (*Lupiáñez et al., 2015*) and often occur in cancer (*Ji et al., 2016*). Disruption of specific, individual CAC-mediated loop anchors using genomic editing tools results in aberrant chromatin contacts and misregulation of neighboring genes in a largely predictable manner, although some redundancy may occur when multiple nearby anchors are present (*Dowen et al., 2014*; *Hnisz et al., 2016*; *Willi et al., 2017*).

The computational method for intra-TAD loop discovery, described here, is a substantial improvement over prior implementations of computational loop prediction (*Sanborn et al., 2015*; *Oti et al., 2016*). The loops we identified were longer and fewer in number (~9500 vs ~60,000), showed much stronger insulation of chromatin interactions and greater insulation of repressive histone marks, and displayed considerably greater overlap with cohesin-mediated loops identified by ChIA-PET using antibody to the cohesin subunit Smc1. Key features of our computational method include the consideration of both CTCF and cohesin binding strength, as noted above, as well as TAD structure and consistency across biological replicates. Our use of both CTCF and cohesin binding strength in predicting intra-TAD loops is supported by a recent study of CTCF sites nearby the mouse α-globin gene cluster, where the presence of CTCF alone was not sufficient to predict DNA loop interactions, and where insulation by individual CAC sites ranged widely – from none to moderate to very strong insulation – in direct proportion to the strength of CTCF binding, as revealed by deletion of individual CTCF sites (*Hanssen et al., 2017*). Furthermore, we developed a simple extension of our method that predicts TAD anchors when given a set of TAD boundaries (*Supplementary file 1C*), and thereby overcome the limitation in identifying TAD anchors from low resolution, standard sequencing depth Hi-C datasets. We were thus able to identify well-defined inter-TAD regions, which we found are enriched for unique gene ontologies, notably, housekeeping genes with ribosomal, nucleosome, and mitosis-related functions. A further extension of our findings would be the explicit use of the intra-TAD and refined-TAD loop coordinates defined here to improve gene target assignments for distal regulatory elements, based on the insulating capacity of these CAC-anchored looped domains. Such an approach may be beneficial for the many model systems where distal enhancer activity is the clear driver of tissue specificity or a given disease state (*Hnisz et al., 2016*). The ability to identify intra-TAD loops based solely on CTCF motif orientation and CTCF and cohesin ChIP-seq binding data, and then use these loops to improve gene target assignments for distal regulatory elements is likely to constitute a substantial improvement over 'nearest gene' and other, more nuanced target assignment algorithms, such as GREAT (*Raviram et al., 2016*).

In conclusion, our studies reveal that while TAD structures are readily apparent in routine Hi-C experiments, their structural organization and functional impact on the genome is not unique. Structurally, the 9,543 TAD-internal sub-loops that we identified for mouse liver have strong cohesin-and-CTCF-bound anchors and appear to be formed by the same loop extrusion mechanism responsible for TAD formation. Functionally, we hypothesize that these intra-TAD loops contribute to nuclear architecture as intra-TAD scaffolds that further constrain enhancer-promoter interactions. We further show that these intra-TAD loops maintain key properties of TADs, most notably insulation of chromatin interactions and insulation of repressive histone mark spreading. The insulation provided by intra-TAD loops may enable high expression of super-enhancer target genes, as illustrated for *Alb* in mouse liver, as well as high expression of individual genes within otherwise inactive TADs, as exemplified by *Scd1* and the many other single gene intra-TAD loops that we identified. Given the increasing interest in interactions of genes with distal enhancers and other intergenic sequences, the rapid and cost-effective method described here for identification of intra-TAD structures that constrain long-range chromatin interactions may prove invaluable in many areas of genomic research.

## Materials and methods

### Animals and processing of liver

Adult male and female CD-1 mice (ICR strain; RRID:MGI:5659424) were purchased from Charles River Laboratories (Wilmington, MA) and were housed in the Boston University Laboratory Animal Care Facility. Animals were treated using protocols specifically reviewed for ethics and approved by Boston University's Institutional Animal Care and Use Committee (IACUC; protocol 16–003). Livers were collected from 8-week-old mice euthanized by cervical dislocation and rinsed in cold PBS. Livers were homogenized in a Potter-Elvehjem homogenizer using high sucrose homogenization buffer (10 mM HEPES (pH 7.5), 25 mM KCl, 1 mM EDTA, 2 M sucrose, 10% glycerol, 0.05 mM DTT, 1 mM PMSF, 0.15 mM spermine, 0.2% (v/v) spermidine, 1 mM Na orthovanadate, 10 mM NaF, and Roche Complete Protease Inhibitor Cocktail) to prevent aggregation of nuclei and preserve chromatin structure. The resulting slurry was transferred on top of a 3 ml cushion of homogenization buffer followed by ultracentrifugation at 25,000 RPM for 30 min at 4°C in an SW41 Ti rotor to pellet the nuclei and remove cellular debris. The supernatant was carefully decanted to remove liquid, and residual solid debris was removed from the tube walls using a sterile spatula and a dampened Kimwipe. Nuclei were resuspended in 1 ml of crosslinking buffer (10 mM HEPES buffer (pH 7.6), 25 mM KCl, 0.15 mM 2-mercaptoethanol, 0.34 M sucrose, 2 mM MgCl$_2$) and transferred to a 1.5 ml Eppendorf tube. To ensure consistent crosslinking, tubes were incubated for 3 min at 30°C prior to the addition of formaldehyde to a final concentration of 0.8% (v/v). Samples were incubated in a 30°C water bath for 9 min with periodic mixing. Crosslinking was halted by the addition of 110 µl of 1 M glycine, followed by a 5 min incubation at room temperature. The crosslinked material was layered on top of 3 ml of high sucrose homogenization buffer and then centrifuged as above. The crosslinked nuclear pellet was resuspended at 4°C in 1 ml of 1X Radioimmunoprecipitation assay (RIPA) buffer (50 mM Tris-HCl, pH 8.0, 150 mM NaCl, 1% IPEGAL, 0.5% deoxycholic acid) containing 0.5% SDS and protease inhibitors until homogenous by gentle pipetting.

### Sonication

Crosslinked nuclei in RIPA buffer containing 0.5% SDS were transferred to 15 ml polystyrene tubes (BD Falcon # 352095) for sonication using a Bioruptor Twin instrument (UCD-400) according to the manufacturer's instructions. Briefly, samples were sonicated at 4°C for 30 s ON and 30 s OFF at high intensity for a total of 75 cycles. Sonicated material was transferred to 1.5 ml Eppendorf tubes, and large debris was cleared by centrifugation at 18,000 x g for 10 min at 4°C. The bulk of this material was snap frozen in liquid nitrogen and stored at −80°C for immunoprecipitation, except that a small aliquot (15 µl) was removed to quantify material and ensure quality by gel electrophoresis, as follows. Aliquots from each sample were adjusted to 0.2 M NaCl, final concentration, then incubated for 6 hr at 65°C. After a three-fold dilution in nuclease-free water, 5 µg of RNase A (Novagen: #70856) was added and samples were incubated for 30 min at 37°C. Samples were then incubated for 2 hr at 56°C with 20 µg of Proteinase K (Bioline; BIO-37084). This material was then quantified in a dilution series using PicoGreen assay (Quanti-iT dsDNA Assay Kit, broad range, Invitrogen) and analyzed on a 1% agarose gel to ensure the bulk of material was within 100–400 bp.

### Chromatin immunoprecipitation (ChIP)

Immunoprecipitation of crosslinked, sonicated mouse liver chromatin and downstream steps were as described previously (*Sugathan and Waxman, 2013*). Protein A Dynabeads (30 µl; Invitrogen: 1002D) were incubated in blocking solution (0.5% bovine serum albumin in PBS) with 5 µl of antibody to CTCF (Millipore #07–729; RRID:AB_441965) or to the cohesin subunit Rad21 (Abcam #992; RRID:AB_2176601) for 3 hr at 4°C. As a control, 1 µl of non-specific rabbit IgG was used (Santa Cruz: sc-2027). Bead immune-complexes were washed with blocking solution, followed by overnight incubation with 70 µg of liver chromatin. After washing with 1X RIPA (containing 0.1% SDS) and reverse crosslinking as described above, DNA was purified using the QIAquick Gel Extraction Kit (Qiagen #28706) and quantified on a Qubit instrument with high sensitivity detection (Invitrogen DNA HS# Q32854), with ChIP yields ranging from 1 to 25 ng. Samples were validated by qPCR using primers shown in *Supplementary file 4B*.

## Library preparation and sequencing

ChIP libraries were prepared for sequencing using NEBNext Ultra II DNA Library Prep Kit for Illumina according to the manufacturer's instructions (NEB, cat. #E7645). All samples were subjected to double-sided SPRI size selection prior to PCR amplification (Agencourt AMPure XP; Beckman Coulter: A63882). Samples were assigned unique barcodes for multiplexing, and subjected to 8 rounds of PCR amplification with barcoded primers (NEB, cat. #E7335). Samples were sequenced either on an Illumina Hi-Seq 4000 instrument at the Duke Sequencing Core or an Illumina Hi-Seq 2000 instrument at the MIT BioMicroCenter, giving 50 bp single end reads at a depth of ~11–19 million reads per sample. A total of four CTCF and three Rad21 (cohesin) ChIP-seq samples were analyzed, representing four male mouse livers. The fourth liver CTCF sequencing sample, sample G133_M9, did not have a matching cohesin ChIP-seq dataset from the same liver sample, and was therefore matched to a merged sample comprised of all three cohesin ChIP-seq replicates (merged at the fastq file level, with processing described below). Raw and processed sequencing data are available at GEO accession number GSE102997.

## General ChIP-seq analysis pipeline

Sequence reads were demultiplexed and mapped to the mouse genome (build mm9) using Bowtie2 (version 2.2.9), allowing only uniquely mapped reads. Peaks of sequencing reads were identified using MACS2 (version 2.1.1) as regions of high signal over background. Peaks were filtered to remove blacklisted genomic regions (www.sites.google.com/site/anshulkundaje/projects/blacklists). Genomic regions called as peaks that contain only PCR duplicated reads, defined as >5 identical sequence reads that do not overlap any other reads, were also removed. All BigWig tracks for visualization in a genome browser were normalized for sequencing depth, expressed as reads per million mapped reads (RPM) using Deeptools (version 2.3.3). Unless otherwise indicated, all pairwise comparisons presented in the Figures were performed using a Kolmogorov–Smirnov test, where **** indicates p≤0.0001, ***p≤0.001, **p≤0.01, and *p≤0.05.

## Motif analysis

Motifs within CTCF peak regions were identified using MEME Suite (version 4.10.0; FIMO and MEME-ChIP options). FIMO was used to assign CTCF motif orientation and motif scores for CAC sites and to discover individual motif occurrences. De novo motif discovery was carried out using MEME-ChIP using default parameters (*Figure 2—figure supplement 3C,D*). Similar results were obtained using Homer (version 4.8). Alternative CTCF motifs were downloaded from CTCFBSDB 2.0 (http://insulatordb.uthsc.edu/download/CTCFBSDB_PWM.mat), however, these did not substantially change any results performed using the core JASPAR motif (MA0139.1). These motifs were explicitly used in *Figure 2—figure supplement 3D*, where no difference between intra-TAD loop anchor and TAD anchor motif usage was observed.

## 4C-seq protocol

Four male and four female mouse livers were processed for *Albumin*-anchored 4C-seq analysis using published protocols, with some changes for primary tissue (*van de Werken et al., 2012*). To adapt the protocol for liver, care was taken to rapidly isolate single liver cells or nuclei suspensions prior to crosslinking. Specifically, two approaches to crosslinking were taken and both gave similar results. One male mouse liver and one female mouse liver sample were processed through the crosslinking step as described for the ChIP protocol, above, prior to quantification of nuclei. The other liver samples (3 males and 3 females) were crosslinked as follows. Half of a liver (~0.5 g) was dissected from each mouse, the gall bladder was removed, and the liver was rinsed with PBS. The liver was then minced and rapidly processed with 10 strokes in a Dounce homogenizer in PBS containing protease inhibitors (PBS-PI; 1X Roche Complete Protease Inhibitor Cocktail; Roche #11697498001). The resulting slurry was passed through a 40-micron cell strainer (Corning #431750), then pelleted and rinsed with PBS-PI (centrifugation at 1,300 RPM for 5 min at 4℃). Following an additional spin, the cell pellet was resuspended well in 9 ml of PBS-PI at room temperature. 270 µl of 37% formaldehyde was added to give a final concentration of 1%, and crosslinking was carried out for 10 min with nutation at room temperature. The remaining formaldehyde was quenched with 1.25 ml of 1 M glycine. Crosslinked cells were pelleted and rinsed with PBS twice (as above) prior to lysis. The supernatant

was removed following the second wash, and cell pellets were resuspended in 8 ml of lysis buffer (50 mM Tris (pH 7.5), 150 mM NaCl, 5 mM EDTA, 0.5% NP-40, 1% TX-100, and 1X Complete Protease Inhibitor Cocktail) and incubated on ice for 40 min with occasional mixing. Lysed cells were spun down at 2,000 RPM for 5 min at 4°C then washed twice with PBS-PI, as above. Nuclei were pelleted, quantified using an Invitrogen Countess instrument, and snap frozen in 10 million nuclei aliquots. Primary digestion of 10 million nuclei with 50,000 U of DpnII (NEB: #R0543) was performed overnight at 37°C in 450 µl of NEBuffer 3 (NEB: #B7003S; 100 mM NaCl, 50 mM Tris-HCl, 10 mM MgCl$_2$, 1 mM DTT, pH 7.9) with agitation at 900 RPM. After confirming digestion by agarose gel electrophoresis, DpnII was inactivated with SDS (2%, final concentration) and the samples then diluted 5-fold in 1X ligation buffer (Enzymatics #B6030). 200 U of T4 DNA ligase was added and primary ligation was carried out overnight at 16°C (Enzymatics #L6030). Ligation was confirmed by analysis of a small aliquot on an agarose gel, and reverse crosslinking was conducted by overnight incubation with 600 µg proteinase K at 65°C. After RNase A digestion and phenol/chloroform cleanup, samples underwent secondary digestion with 50 U of Csp6I (Fermentas #ER0211) overnight at 37°C in 500 µl of 1X Buffer B (Fermentas: #BB5; 10 mM Tris-HCl (pH 7.5), 10 mM MgCl$_2$, 0.1 mg/ml BSA). Csp6I was then heat inactivated for 30 min at 65°C. Samples were diluted 10-fold and secondary ligation was carried out as above, overnight at 16°C. The final PCR template was purified by phenol/chloroform clean up, followed by QiaPrep 2.0 column cleanup (Qiagen #27115). PCR reactions were performed using inversely-oriented 4C primers specific to the *Alb* promoter (sequences shown in bold, below) with dangling 5' half adapter sequences (Reading primer: ACACTCTTTCCCTACACGACGCTCTTCCGA TCT**GGTAAGTATGGTTAATGATC**; Non-reading primer: GACTGGAGTTCAGACGTGTGCTC TTCCGATCT**CTCTTTGTCTCCCATTTGAG**). This design has two advantages: 1) the addition of barcodes in a secondary reaction allows a primer to be reused across samples; and 2) it avoids barcoding at the start of 4C read, which would reduce the mappable read length available for downstream analyses. 4C templates were amplified using Platinum Taq DNA polymerase (Invitrogen #10966026) under the following conditions: 94°C for 2 min, 25 cycles at (94°C 30 s, 55°C 30 s, 72°C 3 min), then 4°C hold. A total of eight liver samples were analyzed (four males, M1-M4; and four females, F1-F4). For liver samples M3, M4, F3 and F4, eight identical PCR reactions for each liver, processed in parallel, were prepared and then pooled to limit the impact of PCR domination in any single reaction. For liver samples M1, M2, F1 and F2, two PCR reactions for each liver were sequenced separately then pooled at the fastq file level for downstream analyses. We observed that the 8 PCR pool reactions gave a more reproducible profile than the single PCR reactions. After pooling, 4C samples were purified using AMPure XP beads (Beckman Coulter: #A63882) at a 1.5:1 ratio of beads to sample, washed with 75% ethanol, dried, and resuspended in 0.1X TE buffer to elute the DNA. 4C-seq samples were multiplexed and PCR amplified using standard NEB barcoded primers (NEB #E7335), as was done for ChIP-seq library preparations, but for five additional PCR cycles rather than the eight cycles used for ChIP libraries (total of 30 cycles of PCR per sample: 25 cycles with viewpoint-specific primers followed by five cycles with viewpoint-generic barcoded NEB primers). 4C libraries were sequenced on an Illumina Hi-Seq 2500 instrument at the New York Genome Center giving 125 bp long paired end reads. Samples were each sequenced to a depth of ~1–5 million reads. Raw and processed sequencing data are available at GEO under accession number: GSE102998.

## 4C-seq data analysis

All *Alb* viewpoint 4C-seq reads were filtered to ensure a match for the bait primers used, then trimmed using a custom script to remove the first 20 nt of each read (*Source code 1*). Reads were then mapped to the mouse mm9 reference genome using the Burrows-Wheeler aligner (bwa-mem) allowing for up to two mismatches. The package r3Cseq (*Thongjuea et al., 2013*) was used to analyze the distribution of signal in *cis*, both near the bait and along chromosome 5. Reads were counted per restriction fragment to obtain the highest possible resolution. Data shown in the main text figures are for the intersection of four male and four female mouse livers, merged according to sex using the intersection option, meaning that the 4C interactions shown are those present in all four samples for a given sex. This produces both a normalized read depth signal (reads per million per restriction fragment) and an associated p-value for the interaction, taking into account distance from viewpoint and reproducibility across replicates (*Figure 6—figure supplement 1A,C*). A comprehensive view of all replicates is presented in *Figure 6—figure supplement 1A*. For all pairwise comparisons, a correlation between samples was calculated genome-wide using the UCSC utility

bigWigCorrelate with default settings. To account for high signal immediately surrounding the viewpoint, this analysis was only conducted for regions > 10 kb from the viewpoint fragment. To analyze more distal *cis* interactions, we first calculated the normalized 4C signal observed per TAD along chromosome 5, in units of RPKM per TAD. We observed a robust logarithmic decay of signal with increasing distance from the viewpoint TAD (*Figure 6C*; $R^2$ = 0.719). Interacting TADs were defined according to observed over expected 4C signal relative to this background model. Interacting TADs were designated as follows: high, defined as regions with >2 fold enrichment over this background model (observed/expected); medium, defined as 1.5 to 2-fold enrichment; and low, between 1.5-fold enrichment and 1.5-fold depletion. Non-interacting TADs showed >1.5 fold depletion of signal. For the *Alb* viewpoint, we identified 17 high, 20 medium, 128 low, and 30 non interacting *cis* TADs along chromosome 5. We required a more comprehensive background model to analyze interactions in *trans*. The tool 4Cker was used for its adaptive windowing and Hidden Markov model approach (*Raviram et al., 2016*). The count tables from r3C-seq were merged by sex and imported, then *trans* analysis was conducted with the recommended parameters (k = 20). The default output identifies three classes of regions: interacting, low-interacting, and non-interacting. For our analysis, the interacting group was divided into two equal-number groups: high-interacting and medium-interacting, based on 4C interaction strength in the male liver samples. *Trans* interacting regions tend to be large (median size of 1.8 Mb), therefore *trans* interacting TADs were defined as TADs wholly contained within these interacting regions. This corresponds to a total of 659 (high), 618 (medium), 969 (low), and 77 (non) interacting *trans* TADs genome-wide.

## CAC sites and scores

Cohesin-and-CTCF (CAC) sites were defined as CTCF peaks that were present in at least 2 of 4 individual mouse liver samples and that overlapped with a cohesin peak in any liver sample. CAC sites were scanned for a CTCF motif (JASPAR motif MA0139.1) within the CTCF peak coordinates using the FIMO tool in the MEME Suite (version 4.10.0). For a given CAC site, the highest scoring motif occurrence for the canonical core CTCF motif (MA0139.1) was considered. A (+) strand orientation indicates that the motif is found on the (+) genomic strand (Watson strand). Each CAC site was represented by two different scores: a CTCF score = p * (m/10), where p is the CTCF peak strength (MACS2 score) and m is the CTCF motif score, as determined by FIMO; and a cohesin score = p * (m/10), where p is the cohesin (Rad21) peak strength (MACS2 score) and m is the CTCF motif score, as determined by FIMO.

## Intra-TAD loop prediction method

We modified a published algorithm for CTCF-mediated loop prediction (*Oti et al., 2016*) to predict intra-TAD loop structures. Key modifications to the algorithm include the following: incorporation of cohesin ChIP-seq data in scoring, based on the finding that CTCF signal in the absence of cohesin is not sufficient to predict chromatin interactions (*Hanssen et al., 2017*); consideration of TAD structure, TSS overlap, and consistency across biological replicates when filtering to obtain the final set of predicted loops; and a final target set of approximately 10,000 intra-TAD loops, based on experimental results from high resolution Hi-C analyses (*Rao et al., 2014*). First, CAC sites were identified from mouse liver ChIP-seq data for Rad21 and CTCF, obtained as described above. Next, each chromosome was scanned for putative intra-TAD loops, formed between a (+) anchor [upstream anchor, that is, CAC site with a CTCF motif (JASPAR motif MA0139.1) on the (+) strand] at the start of a loop and a (-) anchor [downstream anchor, that is, CAC site with a CTCF motif on the (-) strand] at the end of a loop, as described for prediction of intra-chromosomal CTCF loops in (*Oti et al., 2016*). Scanning was initiated from the beginning of each chromosome, and a list of putative (+) anchors was generated. Next: (1) if the next CAC site encountered was a (-) anchor, the pair of (+) and (-) anchors was recorded as a putative intra-TAD loop. The (+) anchor was paired with all subsequent, downstream (-) anchors until another (+) anchor was encountered, at which point the list of putative intra-TAD loops was closed, ending with the last (-) anchor. Alternatively, (2) if the next CAC sites encountered were (+) anchors, then all such (+) anchors were retained as putative upstream anchors, until the next (-) anchor was reached, and then all such (+) anchors were paired (i.e., assigned to loops) with all of the subsequent, downstream (-) anchors until a new (+) anchor was encountered, as described under (1), at which point the list of putative loops was closed, ending with the last (-)

anchor. A new scan for putative intra-TAD loops was then initiated in a linear fashion, starting from the next (+) anchor until all chromosomes were scanned and a set of putative intra-TAD loops was obtained. Chromosome scanning for putative intra-TAD loops was then repeated as described above after removing 10% of the CAC sites – those with the lowest CTCF scores (defined above). Chromosome rescanning was repeated iteratively until the number of putative intra-TAD loops decreased to as close to 20,000 as possible (removing the lowest scoring loops if needed so that all replicates had exactly 20,000 loops prior to merging). A parallel series of iterative scans was carried out, except that 10% of the CAC sites with the lowest cohesin scores (defined above) were removed at each iteration, to generate a second set of ~20,000 putative intra-TAD loops. The intersection of the two sets of 20,000 putative intra-TAD loops was then determined. The same iterative process of intra-TAD loop prediction was carried out independently for each of the n = 4 individual mouse livers, based on an analysis of matched CTCF and cohesin (i.e., Rad21) ChIP-seq datasets for each liver. Thus, for each liver sample, a single putative intra-TAD loop set was generated from the intersection of two sets of predicted CAC-based loops, one using the CTCF score and the other using the cohesin (Rad21) score; these scores were calculated using MACS2 scores for CTCF and cohesin (Rad21), respectively, together with the CTCF motif score m value, as described above. The overlap of these two putative intra-TAD loop sets was approximately 80%, and ranged from 15,999 to 16,892 loops for a given liver sample. Additional filters were then applied to remove intra-TADs that did not contain either a protein-coding TSS or a liver-expressed multi-exonic lncRNA TSS (as defined in [*Melia et al., 2016*]), as we were primarily interested in the impact of intra-TADs on gene expression and regulation. Putative intra-TAD loops that overlapped >80% of the length of a TAD, or whose (+) and (-) anchors are both TAD anchors (defined below) were also excluded, as these loops could not be distinguished from TAD loops. These two filters further reduced the putative intra-TAD loop sets to approximately 63% of the original 20,000 (ranging from 12,395 to 12,962 loops across the four liver samples). A single merged ChIP-seq dataset (merged at the fastq file level, separately for CTCF and for Rad21 datasets) was treated as a fifth dataset. It was run through the full pipeline, above, and then sequentially intersected with the set of putative intra-TAD loops predicted for each individual liver to obtain a final set of 9543 intra-TAD loops identified in all four livers and also present in the 5th dataset (merged sample). Each intra-TAD loop was assigned an intra-TAD loop score equal to the geometric mean of the (+) anchor's CAC site CTCF score and that of its (-) anchor. A second intra-TAD loop score, equal to the geometric mean of the (+) anchor's CAC site cohesin score and that of its (-) anchor, was also assigned. The CTCF and cohesin scores reported for each loop in the final intra-TAD loop lists (*Supplementary file 1B*) are those obtained from the merged sample. Custom scripts for intra-TAD loop prediction are available in *Source code 1*.

Loop predictions for two other mouse cell types (mESC and NPC) and in two human cell lines (GM12878 and K562) was carried out as described above, with the following modifications during filtering. For the mouse cells, ChIP-seq data from biological replicates (n = 4 for mESC and n = 3 for NPC) was obtained from public sources (see below) for CTCF and cohesin (ChIP for the subunit Smc1). Further, TADs from the same cell type were used to filter based on TAD overlap (using TAD boundaries from [*Bonev et al., 2017*]). TSS overlap used the same definitions as above (RefSeq and multi-exonic lncRNA TSS defined in mouse liver [*Melia et al., 2016*]). For human loop predictions, cohesin (ChIP for the subunit Rad21) and CTCF ChIP-seq data were obtained from biological replicates for K562 cells and for GM12878 cells (n = 5 for both cell lines), and overlap with Refgene TSS (hg19) was used to filter the merged loops, as TADs were not defined in (*Rao et al., 2014*). *Supplementary file 4A* provides further details on data sources and accession numbers.

## RNA-seq analysis

Gene expression values for liver-expressed protein coding genes are $\log_2(FPKM + 1)$ values for adult male mouse liver from (*Melia et al., 2016*). Liver-expressed non-coding genes are expressed in FPKM based on the gene models and expression values from (*Melia et al., 2016*). To express the tissue specificity of a gene's expression across a panel of 21 mouse tissues (including liver), we used Tau, which was shown to be the most robust in a recent study (*Kryuchkova-Mostacci and Robinson-Rechavi, 2017*). Testis was excluded from this analysis because a large proportion of testis-expressed genes are highly tissue specific. For each tissue, the maximum FPKM per gene between the two replicates was used. These FPKM values were log transformed and a Tau value, ranging from 0 to 1, was calculated, where one represents high tissue specificity: $\tau = [ \sum_{i=1}^{n} (1 - y_i) ] / (n - 1)$,

where $y_i = x_i / [ \max_{1 \leq i \leq n}(x_i) ]$, n is the number of tissues, and $x_i$ is the expression of the gene in tissue i.

## General Hi-C processing

Hi-C data was processed using the HiC-Pro package (version 2.7.0) (*Servant et al., 2015*) for mapping and read filtering, followed by Homer (version 4.8) for downstream analyses such as PCA analysis and aggregate contact profiles. Biological replicates were merged to increase read depth. The default Homer background model was used for all datasets, where the expected frequency of interactions takes into account read depth between interacting bins and genomic distance. PCA was conducted using Homer with the command 'runHiCpca.pl -res 10000 -cpu 4 -genome mm9' to generate genome wide eigenvalues at 10 kb resolution. The values changed marginally at 20, 40, or 50 kb, but the sign of the eigenvalue was unaffected, that is, there was no impact on whether a TAD was assigned as A compartment or B compartment.

## Peak distribution within TADs

Published TAD coordinates in mouse liver (*Vietri Rudan et al., 2015*) were converted from mouse genome mm10 to mm9 using liftover with default parameters. Each TAD was then divided into 100 equal-sized bins using the Bedtools command makewindows. Next, these bins were compared to the peak positions of various publicly available ChIP-seq datasets using the Bedtools coverage command, and the number of peaks per bin was counted. This resulted in a string of 100 values for each TAD, representing the number of ChIP-seq peaks per bin, where the first value is the start of the TAD and the last value is the end of the TAD. Conducting this analysis across all TADs yielded a matrix of 3617 rows (one per TAD) x 100 columns (one per bin). To generate the aggregate profiles shown in *Figure 1A–1E*, and in *Figure 1—figure supplement 1B–E*, the sum of each column was taken and then normalized to account for differences in total peak count for the different samples, factors, and chromatin marks analyzed. Normalization was conducted by taking the average of the center five bins (bins 48–52) and dividing the bin sums by this normalizing factor. This allows the y axis to represent bin enrichment relative to the center of the TAD, as shown.

## TAD activity and compartment assignment

TAD boundaries were defined at single nucleotide resolution as the end of one TAD and the start of another (as defined in [*Vietri Rudan et al., 2015*]), thus excluding the start of the first TAD in each chromosome and the end of the last TAD. In contrast, all references to 'TAD anchors' refers to the CTCF sites most likely to be anchoring TAD loops based on distance from the boundary and proper orientation (as described in TAD Anchor Identification, below). Data sources for all ChIP-seq, GRO-seq, Hi-C, and other datasets are described in *Supplementary file 4*. H3K9me3, H3K27me3, H2AK5ac, and H3K36me3 marks were processed from the raw sequencing data (fastq files) through the standard ChIP-seq pipeline, above. H3K9me3 and H3K27me3 mark data were expressed as $\log_2$(ChIP/IgG signal). Lamina-associated domain coordinates and GRO-seq data were downloaded as pre-processed data. Heat maps were generated using Deeptools reference point, with a bin size of 10 kb. TAD boundaries were grouped according to k-means clustering (k = 4) using signal within a 1 Mb window from three datasets: H3K9me3, H2AK5ac, and the eigenvalue of the Hi-C PCA analysis (above). Based on these clusters, TADs were classified as active, weak active, weak inactive, or inactive, as follows. If both the start and end boundary of a given TAD were classified as active, then the TAD was designated active. Specifically, a TAD was considered 'active' if the boundary at the start of a TAD fell into clusters 1 or 2 (as marked in *Figure 1F*) and the boundary at the end of the same TAD fell into clusters 1 or 3. The corresponding metric was applied to identify inactive TADs. If the activity status of the start and the end of a given TAD were not in agreement, then the TAD was designated weakly active if the median Hi-C PC1 eigenvalue within the TAD was positive, or weakly inactive if the median Hi-C PC1 eigenvalue was negative. Gene expression and tissue specificity metrics represent expression or Tau values of genes whose TSS overlap active or inactive TADs.

## Additional Hi-C analysis

Contact profiles around TAD, intra-TAD loop, and non-loop-anchor CTCF sites were generated using Homer (v4.9) using the command analyzeHiC and the options '-size 500000 -hist 5000' to

generate interaction profiles for 1 Mb windows around CTCF sites with 5 kb resolution. TAD and intra-TAD loop anchors were split into left and right anchors when found at the start and at the end of the predicted loop, respectively. Non-anchor CTCF sites were defined as other CTCF sites, based on the merged CTCF sample, that also contained a CTCF motif. Left and right groupings were determined based on the orientation of the strongest CTCF motif within the non-anchor peak regions. The inward bias index (IBI) was modified from the more genome-wide directionality index (DI) described in (*Dixon et al., 2012*). Both DI and IBI use a chi-squared statistic to determine the extent to which Hi-C reads from a given region have a strong upstream or strong downstream bias. While DI is genome wide, IBI focuses on the directionality of *cis* interactions (within 2 Mb) from a 25 kb window immediately downstream of a CTCF peak relative to the motif orientation. A large positive value indicates a strong interaction bias towards the loop center, as the motif orientation would predict. Values close to zero indicate a roughly equal distribution of interactions upstream and downstream. By orienting the sign of the IBI value relative to the CTCF motif directionality, we were able to group left and right loop anchors together.

Virtual 4C plots (*Figure 3B* and *Figure 3—figure supplement 1*) and Hi-C screenshots (*Figure 2— figure supplement 5A–CF*, and *Figure 2—figure supplement 7A–C*) were generated using the 3D Genome Browser (http://promoter.bx.psu.edu/hi-c/index.html). Virtual 4C plots used mESC Hi-C with 10 kb resolution and a 25 kb viewpoint for ±250 kb of the selected region. Screenshots were generated for mouse (mESC, CH12, and NPC) and human (GM12878 and K562) cells using raw signal and 10 kb resolution. All Hi-C datasets used were publicly available for mouse (*Rao et al., 2014*; *Bonev et al., 2017*) and human cells (*Rao et al., 2014*) (*Supplementary file 4*).

## TAD anchor identification

TAD anchors were predicted for mouse liver using a modified version of the intra-TAD loop prediction algorithm. The merged list of CTCF peaks was filtered to only consider peaks that were found across all four biological replicates, that contained CTCF motifs, and that were within 50 kb of a TAD boundary, as defined previously for mouse liver (*Vietri Rudan et al., 2015*). This 50 kb distance was chosen based on the ambiguity of binned Hi-C data to more accurately determine the precise TAD boundary. Then, for each TAD boundary, all pairs of (+) and (-) CTCF peaks were considered and scored based on their combined distance to the called TAD boundary. Pairs of '+/-" CTCF peaks that were comprised of a (+) anchor upstream of a (-) anchor (i.e., CTCF peak pairs that were not divergently oriented) were considered an invalid combination to define the end of one TAD and the beginning of the next TAD, and were not considered. The valid pairs with the shortest combined distance to the previously defined liver TAD boundary (*Vietri Rudan et al., 2015*) were retained and all others were removed. If no valid pair for a TAD boundary was identified, the single CTCF peak closest to the TSD boundary was retained as the TAD anchor. A complete listing of TAD anchors is found in *Supplementary file 1C, and a* listing of inter-TAD regions and associated gene ontology analysis is presented in *Supplementary file 3*.

## Alternative loop anchor analysis

We sought to compare the relative insulation of loops identified by our computational approach to alternative loops identified using the original core algorithm of (*Oti et al., 2016*). This provides an objective measure to compare the performance of each computational method in identifying TAD-like loops and loop anchors within TADs. To this end, we used the complete mouse liver CTCF peak list from the merged CTCF sample as input and implemented the loop prediction algorithm exactly as described previously (60% proportional peak cutoff, CTCF signal +motif scores as above, retaining only loops < 200 kb) (*Oti et al., 2016*). As summarized in *Figure 2—figure supplement 1B*, this analysis yielded many more loops (60,678; '60 k loop set') than we obtained using our method (9543 intra-TAD loops). Furthermore, the loops in the 60 k loop set were shorter (median size of 61 kb), and they showed less overlap with cohesin-mediated loops present in the mESC ChIA-PET dataset (25.5% versus 63.2% overlap for our set of intra-TAD loops). 59% of the intra-TAD loops characterized in our study are found in the 60 k loop set. To characterize loops unique to the 60 k loop set, we had to first filter out anchors found intra-TAD or TAD loops (to insure that each list was mutually exclusive, as above). Any 60 k loop anchor within 50 kb of a TAD boundary was excluded from downstream analysis. We also excluded any 60 k loop with at least one anchor that coincided with

an intra-TAD loop anchor. These mutually exclusive lists of intra-TAD loop anchors and the filtered set of 60 k loop anchors (25,983 loop anchors in total; representing a subset of 'Non Anchor CTCF' in the main text, and referred to as '26 k loop anchors' in *Figure 2—figure supplement 1C,D*) were then compared based on insulation of repressive histone marks (see 'Repressive histone mark insulation', below) and Hi-C interaction profiles (see 'Additional Hi-C analysis', above). *Figure 2—figure supplement 1B,C* compares the insular features of intra-TAD loop anchors to those of the set of 26 k alternative loop anchors, which are not intra-TAD loop or TAD anchors.

## Anchor/Loop overlap

CTCF ChIP-seq data for 15 non-liver tissues from the ENCODE Project were downloaded (https://genome.ucsc.edu/cgi-bin/hgTrackUi?db=mm9&g=wgEncodeLicrTfbs) and intersected with replicates to form a single peak list for each tissue (*Shen et al., 2012*). These single peak lists per tissue were then compared to liver CTCF peaks using the Bedtools multiinter command with the –cluster option to generate a union CTCF peak list for all tissues with a score representing the number of tissues in which a peak is present. 'Lone' CTCF (CTCF sites lacking cohesin bound), other/non-anchor CAC sites, TAD anchors, and intra-TAD loop anchors were compared to this list to generate the histograms in *Figure 2C* (see also, *Supplementary file 1C*). Knockdown-resistant cohesin binding sites in liver were defined as Rad21 ChIP-seq peaks found in both wild-type (WT) and Rad21$^{+/-}$ mouse liver, with knockdown-sensitive sites defined as Rad21 peaks found in WT liver that are absent in Rad21$^{+/-}$ liver (*Faure et al., 2012*). Similarly, knockdown-resistant cohesin binding sites in MEFs were defined as Smc1a ChIP-seq peaks present in both WT (*Kagey et al., 2010*) and Stag1-knockout MEFs (*Remeseiro et al., 2012*). Knockdown-sensitive sites were defined as Smc1a peaks found in WT MEFs that are absent in Stag1-knockout MEFs. Phastcons 30-way vertebrate conservation scores were downloaded from the UCSC table browser and converted to BigWig tracks using ucscutils (version 20130327; ftp://hgdownload.cse.ucsc.edu/goldenPath/mm9/phastCons30way/vertebrate). Comparisons to mESC Smc1a ChIA-PET and Smc1a Hi-ChIP datasets (*Dowen et al., 2014*; *Mumbach et al., 2016*) were based on merged replicates, and reciprocal overlaps with intra-TAD loops were required (Bedtools intersect –wa –u –r –f 0.8 –a intraTADloops.bed –b mESC.bed). The mESC Smc1 ChIA-PET dataset was filtered to define 'CTCF-CTCF' interactions as those with both anchor regions overlapping CTCF peak present in a minimum of 2 replicates (total of 3). Any remaining interactions were considered as CNC-mediated enhancer-promoter interactions for the analysis shown in *Figure 2—figure supplement 4C*.

## Repressive histone mark insulation

To determine if a TAD or intra-TAD loop anchor CAC showed more insulation, or less insulation, than other classes of CTCF or cohesin binding sites, we used Jensen Shannon Divergence (JSD; [*Fuglede and Topsoe, 2004*]) to quantify the insulation of H3K27me3 and H3K9me3 ChIP-seq signals. Specifically, regions 10 kb upstream and 10 kb downstream of each peak in a given peak list (i. e., TAD anchors, CNC, etc.) were each divided into 50 bins of 200 bp each. The number of H3K27me3, H3K9me3, or IgG ChIP-seq reads within each bin was tallied, resulting in a vector of 50 + 50 values for each peak region. These were then compared to two test vectors representing complete (maximal) insulation: fifty 0's followed by fifty 1's, and fifty 1's followed by fifty 0's. These are theoretical representations of low signal upstream of the peak followed by high signal downstream, and vice versa. Using a custom python script (*Source code 1*), the similarity between the experimentally derived vector and each of the test vectors was calculated, where a lower value represents less divergence from the test vector. The cumulative frequency distribution per group (anchors, CAC, CNC, etc.) is presented for the most similar test vector per peak in *Figure 3D and E* (K27me3 and K9me3) and *Figure 3—figure supplement 2D* (IgG). Heat maps show ChIP signal Z-transformed data across all CTCF-bound regions.

## Five class DHS model

The ~70,000 open chromatin regions (DHS) previously identified in mouse liver (*Ling et al., 2010*) were classified based on ChIp-seq signals for H3K4me1, H3K4me3, and CTCF within 1 kb of each DHS summit, obtained using the refinepeak option in MACS2. The general schematic is shown in *Figure 4—figure supplement 1A*. Promoter DHS were defined as DHS with a > 1.5 fold ratio of

H3K4me3 relative to H3K4me1 ChIP-seq signal; enhancer DHS were defined as DHS with a < 0.67 fold ratio of H3K4me3 relative to H3K4me1 ChIP-seq signal, calculated as reads per million for each factor. Both DHS sets were filtered to remove DHS with <4 reads per million for both marks after subtracting IgG signal (*Figure 4A*). These cutoff values leave two remaining DHS groups, one with a roughly equal ratio between the two histone-H3 marks, and one with low signal (<4 reads per million) for both marks. The former DHS were classified as weak promoter DHS, based on their close proximity to RefSeq TSS and the low expression of neighboring genes (*Figure 4—figure supplement 1B,C*). The remaining DHS group, characterized by low ChIP-seq signals, was largely intergenic but showed weak to undetectable levels of canonical histone marks. Low signal regions that overlapped a CTCF site with higher CTCF ChIP-seq signals than H3K4me1 signals were classified as insulators (*Figure 4—figure supplement 1A*). The remaining regions were designated weak enhancer-DHS based on their distance from TSS and their low levels of H3K27ac ChIP-seq signal compared to the enhancer-DHS group (*Figure 4—figure supplement 1B*). The majority of promoter-DHS and weak promoter-DHS were <1 kb from a TSS (*Figure 4—figure supplement 1B*). To compare the level of expression for genes with promoter-DHS or weak promoter-DHS (*Figure 4—figure supplement 1C*), the TSS was required to be within 10 kb of the DHS summit. Any gene with both a weak promoter-DHS and a promoter-DHS within 10 kb was categorized as being regulated by a promoter-DHS; thus, there was no overlap between weak promoter-DHS-regulated genes and promoter-DHS-regulated genes.

## Comparison of DHS classes across tissues

All available mouse tissue DNase-seq peak regions were downloaded from the ENCODE Project website (https://www.encodeproject.org/) (*Shen et al., 2012*). ENCODE mm9 blacklist regions (https://sites.google.com/site/anshulkundaje/projects/blacklists) were removed, and the lists were merged to form a single reproducible peak list for each tissue, as follows. Due to variable replicate numbers across tissues, the following cutoffs were used to form merged DHS lists for each tissue. If a tissue had only two replicates (as was the case for 12 of the 20 non-liver tissues), we required that the DHS be present in both replicates. If a tissue had 3 or 4 replicates, then the DHS were required to be present in all or all but one replicate (this was the case for 7 of the 20 non-liver tissues). For whole brain tissue, the merged peak list required that a DHS was present in at least 5 of the 7 replicates. These regions were compared to each other using the Bedtools multiinter command with the –cluster option to generate a union DHS peak list for all tissues, where the score column represents the number of non-liver tissues in which a given region was found. For all liver DHS assigned to one of the above five DHS classes (*Supplementary file 2A*), each liver DHS summit was mapped to this all tissue union peak list, allowing only one match per summit up to 150 nt away. If a given liver DHS summit was >150 nt from the nearest DHS in any other tissue, it was given a score of '0' for liver-specificity. Otherwise the score represents the number of mouse tissues that the closest DHS was found in.

## Super-enhancer identification

Super-enhancers were identified using the ROSE (Ranked Order of Super Enhancers) software package (http://younglab.wi.mit.edu/super_enhancer_code.html). ROSE takes a list of enhancer regions and mapped read positions as input to identify highly active clusters of enhancers. Default options were used, including 12.5 kb as the maximum distance for grouping (stitching) enhancers into putative super-enhancers, as well as reads per million normalization for all H3K27ac ChIP signal used for ranking enhancer clusters. The set of all enhancer-DHS and weak enhancer-DHS regions from the five class DHS model described above (*Supplementary file 2A*) was used as the region input list. A set of 19 publicly available H3K27ac ChIP-seq datasets from mouse liver was used as signal input (see *Supplementary file 4* for sample information). This set includes datasets for male, female (*Sugathan and Waxman, 2013*), and circadian time course (male only; [*Koike et al., 2012*]) mouse liver datasets. A strict intersection of super-enhancers identified across all 19 samples was used to define a set of 503 'core' super-enhancers in mouse liver using the Bedtools multiintersect command, as shown in *Figure 4—figure supplement 2A*. Any enhancer cluster (i.e., constituent enhancers within 12.5 kb, as above) not identified as a super-enhancer in any sample was termed a typical enhancer and considered as individual constituents only.

Gene targets for enhancers were assigned as the nearest gene (based on TSS position up to a maximum distance cutoff of 10 or 25 kb, as specified. Gene expression values and tissue specificity were defined as described above. Aggregate plots were generated using Deeptools (version 2.3.3). In *Figure 4F*, the scale-regions option of Deeptools was used to scale super-enhancers and typical enhancers to their median sizes of 44 kb and 1 kb, respectively. *Figure 4—figure supplement 2B* used the reference-point mode of Deeptools and shows GRO-seq signal that overlaps eRNA loci as defined previously (*Fang et al., 2014*). Super-enhancer and typical enhancer coordinates for mESC and ProB cells are from (*Whyte et al., 2013*).

## Data availability

Data generated and used in this study has been deposited in the Gene Expression Omnibus (GEO) under accession number GSE102999 (https://www.ncbi.nlm.nih.gov/geo/query/acc.cgi?acc=GSE102999). ChIP-seq data are available under the subseries GSE102997 (https://www.ncbi.nlm.nih.gov/geo/query/acc.cgi?acc=GSE102997). 4C-seq data are available under the subseries GSE102998 (https://www.ncbi.nlm.nih.gov/geo/query/acc.cgi?acc=GSE102998). Published datasets used in this study are listed in *Supplementary file 4*.

## Acknowledgements

We thank Andy Rampersaud and Dr. Tisha Melia of this laboratory for their work building and standardizing the ChIP-seq and RNA-seq pipelines used in this study. We also thank Aram Shin for first piloting CTCF and Rad21 ChIP-seq experiments in this lab. This work was supported by National Institutes of Health grants [grant numbers DK33765, ES024421 to DJW]; and by the National Science Foundation predoctoral fellowship [grant number DGE-1247312 to BJM]. Funding for open access charge: National Institutes of Health.

## Additional information

### Funding

| Funder | Grant reference number | Author |
| --- | --- | --- |
| National Institutes of Health | DK33765 | David J Waxman |
| National Science Foundation | DGE-1247312 | Bryan J Matthews |
| National Institutes of Health | ES024421 | David J Waxman |

The funders had no role in study design, data collection and interpretation, or the decision to submit the work for publication.

### Author contributions

Bryan J Matthews, Conceptualization, Laboratory data acquisition, Data analysis, Figure preparation, Writing—original draft; David J Waxman, Overall research guidance and project mentorship, Writing—review, revision and editing, Project administration and funding acquisition

### Author ORCIDs

Bryan J Matthews (iD) https://orcid.org/0000-0002-1930-339X
David J Waxman (iD) http://orcid.org/0000-0001-7982-9206

### Ethics

Animal experimentation: Adult male and female CD-1 mice (ICR strain) were purchased from Charles River Laboratories (Wilmington, MA) and were housed in the Boston University Laboratory Animal Care Facility. Animals were treated using protocols specifically reviewed for ethics and approved by Boston University's Institutional Animal Care and Use Committee (IACUC; protocol 16-003).

Decision letter and Author response
Decision letter https://doi.org/10.7554/eLife.34077.070
Author response https://doi.org/10.7554/eLife.34077.071

## Additional files

### Supplementary files

• Source Code 1. All code used for the prediction of intra-TAD loops and additional custom scripts.
DOI: https://doi.org/10.7554/eLife.34077.023

• Supplementary file 1. Intra-TAD loop and CTCF coordinates relevant to *Figures 1–3*.
DOI: https://doi.org/10.7554/eLife.34077.024

• Supplementary file 2. Mouse liver DHS and Super-enhancer information relevant to *Figure 4*.
DOI: https://doi.org/10.7554/eLife.34077.025

• Supplementary file 3. Genes and GO term enrichments for genes at refined TAD boundaries and in intra-TADs.
DOI: https://doi.org/10.7554/eLife.34077.026

• Supplementary file 4. Publicly available datasets used and qPCR primers used to validate CTCF/Rad21 ChIPs.
DOI: https://doi.org/10.7554/eLife.34077.027

• Transparent reporting form
DOI: https://doi.org/10.7554/eLife.34077.028

### Data availability

Data generated and used in this study has been deposited in the Gene Expression Omnibus (GEO) under accession number GSE102999 (https://www.ncbi.nlm.nih.gov/geo/query/acc.cgi?acc=GSE102999). ChIP-seq data are available under the subseries GSE102997 (https://www.ncbi.nlm.nih.gov/geo/query/acc.cgi?acc=GSE102997). 4C-seq data are available under the subseries GSE102998 (https://www.ncbi.nlm.nih.gov/geo/query/acc.cgi?acc=GSE102998). Published datasets used in this study are listed in Table S4.

The following datasets were generated:

| Author(s) | Year | Dataset title | Dataset URL | Database, license, and accessibility information |
| --- | --- | --- | --- | --- |
| Matthews BJ, Waxman DJ | 2018 | 4C-seq analysis of interactions with the Albumin promoter in mouse liver | https://www.ncbi.nlm.nih.gov/geo/query/acc.cgi?acc=GSE102998 | Publicly available at the NCBI Gene Expression Omnibus (accession no: GSE102998) |
| Matthews BJ, Waxman DJ | 2018 | CTCF and Cohesin (Rad21) ChIP-seq in mouse liver | https://www.ncbi.nlm.nih.gov/geo/query/acc.cgi?acc=GSE102997 | Publicly available at the NCBI Gene Expression Omnibus (accession no: GSE102997) |
| Matthews BJ, Waxman DJ | 2018 | Computational prediction of CTCF/cohesin-based intra-TAD (sbTAD) loops that insulate chromatin contacts and gene expression in mouse liver | https://www.ncbi.nlm.nih.gov/geo/query/acc.cgi?acc=GSE102999 | Publicly available at the NCBI Gene Expression Omnibus (accession no: GSE102999) |

The following previously published datasets were used:

| Author(s) | Year | Dataset title | Dataset URL | Database, license, and accessibility information |
| --- | --- | --- | --- | --- |
| Faure AJ, Schmidt D, Watt S, Schwalie PC, Wilson MD, Xu H, Ramsay RG, | 2012 | ChIP-seq in primary wild type and Rad21-cohesin haploinsufficient mouse liver using cohesin, CTCF and a collection of tissue-specific | https://www.ebi.ac.uk/arrayexpress/experiments/E-MTAB-941/ | Publicly available at EBI Array Express (accession no. E-MTAB-941) |

| Author | Year | Title | URL | Availability |
|---|---|---|---|---|
| Odom DT, Flicek P | | and ubiquitous regulators | | |
| Huayun Hou, Michael D Wilson | 2016 | Genomic localization of Topoisomerase II beta | https://www.ebi.ac.uk/arrayexpress/experiments/E-MTAB-3587/ | Publicly available at EBI Array Express (accession no. E-MTAB-3587) |
| Schmidt D, Schwalie PC, Wilson MD, Brown GD, Flicek P, Odom DT | 2012 | ChIP-seq study of CTCF binding evolution in human, macaque, mouse, rat and opossum | https://www.ebi.ac.uk/arrayexpress/experiments/E-MTAB-437 | Publicly available at EBI Array Express (accession no. E-MTAB-437) |
| Sugathan A, Laz EV, Waxman DJ | 2013 | Genome-wide maps of histone modifications in male and female mouse liver | https://www.ncbi.nlm.nih.gov/geo/query/acc.cgi?acc=GSE44571 | Publicly available at the NCBI Gene Expression Omnibus (accession no: GSE44571) |
| Shen Y, Yue F, Ren B | 2012 | A draft map of cis-regulatory sequences in the mouse genome | https://www.ncbi.nlm.nih.gov/geo/query/acc.cgi?acc=GSE29184 | Publicly available at the NCBI Gene Expression Omnibus (accession no: GSE29184) |
| Koike N, Yoo S, Huang H, Kumar V, Lee C, Kim T, Takahashi JS | 2012 | Transcriptional Architecture and Chromatin Landscape of the Core Circadian Clock in Mammals [ChIP-seq] | https://www.ncbi.nlm.nih.gov/geo/query/acc.cgi?acc=GSE39977 | Publicly available at the NCBI Gene Expression Omnibus (accession no: GSE39977) |
| Ling G, Sugathan A, Mazor T, Fraenkel E, Waxman DJ | 2010 | Unbiased, Genome-wide in vivo Mapping of Transcriptional Regulatory Elements Reveals Sex Differences in Chromatin Structure Associated with Sex-specific Liver Gene Expression | https://www.ncbi.nlm.nih.gov/geo/query/acc.cgi?acc=GSE21777 | Publicly available at the NCBI Gene Expression Omnibus (accession no: GSE21777) |
| Yue F, Cheng Y, Breschi A, Vierstra J | 2014 | A comparative encyclopedia of DNA elements in the mouse genome | https://www.ncbi.nlm.nih.gov/geo/query/acc.cgi?acc=GSE49847 | Publicly available at the NCBI Gene Expression Omnibus (accession no: GSE49847) |
| Vietri Rudan M, Barrington C, Ernst C, Odom DT, Tanay A, Hadjur S | 2015 | Comparative Hi-C reveals that CTCF underlies evolution of chromosomal domain architecture | https://www.ncbi.nlm.nih.gov/geo/query/acc.cgi?acc=GSE65126 | Publicly available at the NCBI Gene Expression Omnibus (accession no: GSE65126) |
| Dowen JM, Fan ZP, Hnisz D, Ren G | 2014 | Long-range chromatin interactions in mouse embryonic stem cells identified by ChIA-PET | https://www.ncbi.nlm.nih.gov/geo/query/acc.cgi?acc=GSE57913 | Publicly available at the NCBI Gene Expression Omnibus (accession no: GSE57913) |
| Mumbach MR, Rubin AJ, Flynn RA, Dai C, Khavari PA, Greenleaf WJ, Chang HY | 2016 | HiChIP: Efficient and sensitive analysis of protein-directed genome architecture | https://www.ncbi.nlm.nih.gov/geo/query/acc.cgi?acc=GSE80820 | Publicly available at the NCBI Gene Expression Omnibus (accession no: GSE80820) |
| Hao P, Melia T, Sugathan A, Waxman DJ | 2015 | RNA-seq analysis of gene expression in male and female mouse liver | https://www.ncbi.nlm.nih.gov/geo/query/acc.cgi?acc=GSE48109 | Publicly available at the NCBI Gene Expression Omnibus (accession no: GSE48109) |
| Lin S, Lin Y, Nery JR, Urich MA | 2014 | Long RNA-seq from ENCODE/Cold Spring Harbor Lab | https://www.ncbi.nlm.nih.gov/geo/query/acc.cgi?acc=GSE36025 | Publicly available at the NCBI Gene Expression Omnibus (accession no: GSE36025) |
| Fang B, Everett LJ, Jager J, Briggs E, Armour SM, Feng D, Roy A, Gerhart-Hines Z, Sun Z, Lazar MA | 2014 | Circadian Enhancers Coordinate Multiple Phases of Rhythmic Gene Transcription In Vivo | https://www.ncbi.nlm.nih.gov/geo/query/acc.cgi?acc=GSE59486 | Publicly available at the NCBI Gene Expression Omnibus (accession no: GSE59486) |

| Fu Y, Lv P, Yan G, Fan H | 2015 | Genome-wide maps of nuclear lamina interactions in AML12 cells | https://www.ncbi.nlm.nih.gov/geo/query/acc.cgi?acc=GSE73703 | Publicly available at the NCBI Gene Expression Omnibus (accession no: GSE73703) |
|---|---|---|---|---|
| Hon GC, Rajagopal N, Shen Y | 2013 | Tissue-specific methylomes reveal epigenetic memory in adult mouse tissue | https://www.ncbi.nlm.nih.gov/geo/query/acc.cgi?acc=GSE42836 | Publicly available at the NCBI Gene Expression Omnibus (accession no: GSE42836) |
| Reizel Y, Spiro A, Sabag O, Skversky Y | 2015 | Gender-specific post-natal demethylation and establishment of epigenetic memory | https://www.ncbi.nlm.nih.gov/geo/query/acc.cgi?acc=GSE60012 | Publicly available at the NCBI Gene Expression Omnibus (accession no: GSE60012) |

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
