## [Decision Letter]

Thank you for sending your article entitled "Insulation of gene expression by CTCF and cohesin-based subTAD loop structures" for peer review at *eLife*. Your article has been favorably evaluated by Jessica Tyler (Senior Editor) and three reviewers, one of whom is a member of our Board of Reviewing Editors.

The reviewers overall appreciated the extensive analysis performed by the authors, and recognized the potential impact of this work upon further analysis. A major concern raised by all reviewers is that further evidence is required to better characterize the subTAD predictions made by the authors and relate these to known phenomena. The reviewers suggested using the available high-resolution Hi-C datasets by Rao et al., 2014 and Bonev et al., 2017 to comprehensively evaluate how the subTAD predictions made by the authors relate to small TADs and localized loop interactions which were found in these datasets. If successful, such an analysis will significantly strengthen the applicability of this approach as an alternative for high-resolution Hi-C. This would allow to evaluate whether loop interactions detected in Hi-C are mostly explained by convergent CTCF interaction. Together, these would significantly strengthen the paper.

In addition, the reviewers suggest restructuring the manuscript such that previously known results and anecdotal results are made less central in the manuscript (e.g. by moving these to the supplement).

*Reviewer #1:*

In this work the authors extend a previously-published computational method, that uses ChIP-seq data to predict chromatin loop structures, in order to identify loops within TADs. They refer to these loops as subTADs (confusing terminology, see below). Based on this method and on several published datasets, the authors analyze several aspects of the relations between TADs, subTADs, epigenetic state and gene expression. Their main biological result suggests that subTAD loop anchors are functionally similar to TAD loop anchors.

While the presented method is useful, taken together, in my view both the methodological and biological findings in this manuscript are not of sufficient novelty.

1) The authors define subTADs as intra-TAD loops which are anchored by CTCF and cohesin. This definition is very confusing, since the term TAD typically refers to a domain-like square/triangle interaction pattern in 3C-type experiments, and subTADs are typically defined as TAD-like patterns that are contained within larger TADs (e.g. Phillips-Cremins Cell 2013). It has been observed that TAD patterns often have end-to-end loop patterns (by loops I mean local peaks in Hi-C, as defined by the authors), but these phenomena are generally distinct as some TADs do not show localized looping interactions and some looping interactions appear outside TADs.

2) However, this is more than an issue of terminology, since the subTAD loops detected in this work may actually just represent small TADs. If this is the case, there may not be any reason to expect them to behave differently than other larger TADs, simply because the decision of what is called a TAD vs. subTAD is quite arbitrary in TAD-detection methods. In other words, the finding that TAD anchors are functionally similar to subTAD anchors may be of little novelty if the methodological distinction between the two is not clearly motivated.

3) In terms of methodological novelty, the method used to predict subTAD loops is novel but incremental, as its core has been published (prediction of loops based on CTCF ChIP-seq and motif directionality). The authors extend it to include cohesin occupancy and filter some loops so that they do not overlap TAD loops etc., but this extension is not a conceptual advance in the method (in fact the panel explaining the method is very similar to a panel from the previous paper describing the method).

4) Some of the analyses in the manuscript are of little novelty and are only indirectly linked to subTAD analysis (e.g. Figure 1 is a general analysis of TADs (A-E) and the association between gene expression and chromatin state (F-H); Figure 4 presents an analysis of enhancers and super enhancers, mostly unrelated to TADs or subTADs).

5) As far as I understand, "other CAC" sites will mostly be weak sites that did not pass the filters (this is supported by Figure 2F). How do we even know they are real sites? Given this observation, it may not be meaningful to compare with them.

6) The same goes for "other CTCF" sites: looking at Figure 2F, the ChIP signal shows they are mostly not actual sites.

*Reviewer #2:*

In the manuscript "Insulation of gene expression by CTCF and cohesin-based subTAD loop structures" Mathews and Waxman provide a detailed characterization of their CTCF and cohesin ChIP-seq on mouse liver. They focus on TADs and characterize their features with the help of published datasets. The characterization is later use as a proxy to compare TADs with identified/predicted subTADs. They use a previously published algorithm, to predict TADs, modified by the authors to predict subTADs by considering genomic and structural parameters. Furthermore, they identify enhancer, insulators and super-enhancers specific to mouse liver with the help of previously published data. They end by experimentally testing a subTAD insulation containing a super-enhancer.

I would recommend for publication in *eLife* with major modifications.

1) It is this reviewer's opinion that an essential part missing from this manuscript is for the authors to clearly show examples of subTADs that are not detected by Hi-C or ChIA-PET but are indeed predicted by their computational method. This will help the readers make sense of the very detailed bioinformatics analyses and bring home the relevance of the prediction of subTADs.

2) Why use 4C and not 5C, capture Hi-C or similar approach where you would be able to visualize your subTADs. I understand these approaches cost can be higher yet if not that at least the addition of extra viewpoints inside and outside the subTADs would be needed to show the insulation experimentally. Although unlikely, what is shown here can simply be explained by the 4C interaction frequency dropping exponentially from the viewpoint that is conveniently located at the center of the subTAD.

3) The extent of how the authors modified the method previously published and improved upon is largely unclear. I strongly suggest they make the code/scripts available to the community for all of their analyses otherwise their contributions most likely will remain unnoticed/unused.

4) Concerns about novelty: besides the obvious similarity from approaches published before additionally there is a paper found on bioRxiv (Predicting CTCF-mediated chromatin interactions by integrating genomic and epigenomic features. Kai et al., 2017) that predicts subTADs based on machine learning of ChIA-PET results. Might be helpful to discuss the pros/cons between methods.

*Reviewer #3:*

The authors build on a previous method that uses CTCF binding and orientation (and predicts nearly 60k loops of average length 61kb) to define subTADs that are strictly contained in TADs and are fewer in number than the original method that predicted 60k loops. Motivated by other recent findings they incorporated cohesin ChIP-seq data as well as TAD boundaries, TSS overlap and consistency across replicates and trained their model to come up with a target number of approximately 10k (number of contact domains identified from 1kb/5kb resolution Hi-C data) subTADs with convergent CTCF sites on their boundaries. Even though there is not much in terms of robustness and stability analysis of some selected parameters and targets, their approach is mostly justifiable by the literature and what we know about TADs.

The manuscript thoroughly characterizes the properties of these newly defined subTADs in mouse liver and compares them to TADs and several other types of "domains". Main conclusions are that subTADs are slightly weaker but smaller versions of TADs that carry all key TAD features and provide finer-scale control of gene expression.

1) My one main concern is the missing out on an opportunity to clearly distinguish these subTADs from the contact domains defined by Rao et al., 2014 paper on several different cell lines. All the data needed to perform subTAD predictions are available for most (if not all) of these cell lines and a clear, cell-type matched comparison of what subTADs are and how they overlap with contact domains (as well as the subset of loop domains). Only in Figure S4C there is an indirect comparison of contact domains from a mouse B-cell lymphoma cell line (CH12-LX) and subTADs from mouse liver. I think it is crucial for the field to understand similarities and differences of many different definitions of domains and subdomains in order to converge to a consensus. This paper could have done more towards that goal.

---

## [Author Response]

The reviewers overall appreciated the extensive analysis performed by the authors, and recognized the potential impact of this work upon further analysis. A major concern raised by all reviewers is that further evidence is required to better characterize the subTAD predictions made by the authors and relate these to known phenomena. The reviewers suggested using the available high-resolution Hi-C datasets by Rao et al., 2014 and Bonev et al., 2017 to comprehensively evaluate how the subTAD predictions made by the authors relate to small TADs and localized loop interactions which were found in these datasets. If successful, such an analysis will significantly strengthen the applicability of this approach as an alternative for high-resolution Hi-C. This would allow to evaluate whether loop interactions detected in Hi-C are mostly explained by convergent CTCF interaction. Together, these would significantly strengthen the paper.In addition, the reviewers suggest restructuring the manuscript such that previously known results and anecdotal results are made less central in the manuscript (e.g. by moving these to the supplement).

We have analyzed the Hi-C datasets provided by Rao et al., 2014 and Bonev et al., 2017, and our results support the utility of our method as an alternative to high-resolution Hi-C, as described below in our response to reviewer 3, comment 1. We compare our loop predictions based on convergent CTCF interactions to loop predictions directly from Hi-C and ChIA-PET in both mouse and human cell types. We did not move Figure 1 or Figure 4 to the supplement, based on specific guidance from the Editor on this point.

Reviewer #1:[…] While the presented method is useful, taken together, in my view both the methodological and biological findings in this manuscript are not of sufficient novelty.1) The authors define subTADs as intra-TAD loops which are anchored by CTCF and cohesin. This definition is very confusing, since the term TAD typically refers to a domain-like square/triangle interaction pattern in 3C-type experiments, and subTADs are typically defined as TAD-like patterns that are contained within larger TADs (e.g. Phillips-Cremins Cell 2013). It has been observed that TAD patterns often have end-to-end loop patterns (by loops I mean local peaks in Hi-C, as defined by the authors), but these phenomena are generally distinct as some TADs do not show localized looping interactions and some looping interactions appear outside TADs.

We changed the term used to describe these loops, from “subTAD loops” to “intra-TAD loops”, throughout the manuscript, and in all figures and legends. We now see that our use of the term intra-TAD loops accurately reflects the comparisons that we made to larger looped TAD structures, while eliminating the issues noted by the reviewer.

2) However, this is more than an issue of terminology, since the subTAD loops detected in this work may actually just represent small TADs. If this is the case, there may not be any reason to expect them to behave differently than other larger TADs, simply because the decision of what is called a TAD vs. subTAD is quite arbitrary in TAD-detection methods. In other words, the finding that TAD anchors are functionally similar to subTAD anchors may be of little novelty if the methodological distinction between the two is not clearly motivated.

Yes, our findings do demonstrate that small TAD-like loops are present within larger TAD loops. An important outcome of our study, however, is that these small TAD-like loops have properties consistent with those of the larger TAD loops, and perhaps more importantly, can be reliably predicted by our analysis method, which presents what we believe to be a rather useful alternative to high-resolution Hi-C. We essentially started with the concept that TADs subdivide the genome into functional blocks of genomic activity/inactivity but that gene expression and activity is not uniform within many TADs. Our study constitutes a useful approach to reliably subdivide intra-TAD space into finer-scale blocks of regulatory domains. Our method is most useful when Hi-C data is not available, as it requires ChIP-seq data for only two factors: CTCF and Cohesin.

Prior efforts ignored two key features of our analysis method, cohesin signal strength and TAD structure. Regardless of what nomenclature is best used to describe these intra-TAD loop structures, we believe that the practical application of our method is its ability to identify these structures, which we show impact gene expression and regulation. We had demonstrated the utility of our method for mouse liver, and with the new results added to the revised manuscript, we show that our method can successfully predict loops in other mouse cell types (mESC and NPC) and also in human cells (GM12878 and K562 cells). For mESC and NPC cells, where TADs were defined using the highest resolution Hi-C currently available, our method is able to correctly detect intra-TAD loop structures (supported by ChIA-PET and in the Hi-C contact matrix) that were not detected as “small TADs,” further strengthening the utility of our method.

3) In terms of methodological novelty, the method used to predict subTAD loops is novel but incremental, as its core has been published (prediction of loops based on CTCF ChIP-seq and motif directionality). The authors extend it to include cohesin occupancy and filter some loops so that they do not overlap TAD loops etc., but this extension is not a conceptual advance in the method (in fact the panel explaining the method is very similar to a panel from the previous paper describing the method).

Our computational method is based on a prior published algorithm, as we stated in the manuscript, that we modified to incorporate certain genomic features and structural parameters not considered in the prior method, which was devised for a distinct, albeit related purpose (predicting any putative CTCF interaction vs. predicting intra-TAD interactions). While we can see why the reviewer described our changes as incremental, small changes in approach can sometimes lead to dramatically improved results – as is the case here. Evidence for this improvement is provided, for example, in Figure 2—figure supplement 1, where we present a direct comparison of the results of our method with that of the prior method. One of the key ‘incremental’ changes is the incorporation of cohesin ChIP-seq data into our loop prediction model. Independent support for the importance of cohesin has now been provided in the form of a preprint recently made available by Kai et al. 2017 (also see our response to reviewer 2, comment 4 below).

4) Some of the analyses in the manuscript are of little novelty and are only indirectly linked to subTAD analysis (e.g. Figure 1 is a general analysis of TADs (A-E) and the association between gene expression and chromatin state (F-H); Figure 4 presents an analysis of enhancers and super enhancers, mostly unrelated to TADs or subTADs).

Per the guidance of the senior editor, we did not alter the presentation in Figures 1 and 4, which we consider as narratively important to understand the breadth of intra-TAD loops’ impact on gene expression and enhancer-promoter interaction.

5) As far as I understand, "other CAC" sites will mostly be weak sites that did not pass the filters (this is supported by Figure 2F). How do we even know they are real sites? Given this observation, it may not be meaningful to compare with them.6) The same goes for "other CTCF" sites: looking at Figure 2F, the ChIP signal shows they are mostly not actual sites.

Yes, “Other CAC” and “Lone CTCF” sites tend to be weaker sites, however, we were careful to ensure that there is strong experimental support for all of the sites included in our analyses: each site was identified in both the merged sample and in a minimum of two individual biological replicates. Additionally, in our comparison across mouse tissues presented in Figure 2C, we found that 93% of all the “Other CAC” sites used in our analyses were verified to be bound by CTCF in at least one other mouse tissue, and 66% were verified in at least 6 other tissues. Similarly, for “Lone CTCF”, 81% of sites were bound by CTCF in at least one other mouse tissue, and 39% were verified in at least 6 other tissues. We have revised the manuscript to incorporate these findings about the robustness of the Other CAC and Lone CTCF sites (see legend to Figure 2C). Further, we now mention in the fourth paragraph of the subsection “Intra-TAD loops show strong, directional interactions and insulate chromatin marks” that all CTCF peaks were required to be present in a minimum of 2 individual biological replicates.

Reviewer #2:[…] I would recommend for publication in eLife with major modifications.1) It is this reviewer's opinion that an essential part missing from this manuscript is for the authors to clearly show examples of subTADs that are not detected by Hi-C or ChIA-PET but are indeed predicted by their computational method. This will help the readers make sense of the very detailed bioinformatics analyses and bring home the relevance of the prediction of subTADs.

The reviewer asks what properties or features might characterize subTADs (now referred to as intra-TAD loops; see above) that we predicted computationally in liver but for which there is no supporting experimental data in the available Hi-C or ChIA-PET datasets from other mouse tissues. Overall, we found that ~75% of our predicted intra-TAD loops have support from experimental Hi-C, Hi-ChIP, or ChIA-PET datasets (subsection “Intra-TAD loops in other mouse tissues and in human cells”, second paragraph). To address the reviewer’s comment, we examined the remaining set of ~25% of subTAD (intra-TAD) loops, which are apparently tissue-specific. We also examined two other sets of tissue-specific loops that we identified by applying our algorithm to two other mouse cell types and two human tumor cell lines (Figure 2—figure supplements 3 and 4). Our findings, presented in Figure 2—figure supplements 3C 4C show that tissue-specific loops tend to be weaker (as measured by ChIA-PET interaction strength in mESC and K562 cells), but are real and observable in the Hi-C contact matrices (Figure 2—figure supplements 3E-F and 4E-F). Further, these tissue-specific loops show directional interactions in the new virtual 4C data that we have now added to the manuscript (Figure 3—figure supplement 1).

2) Why use 4C and not 5C, capture Hi-C or similar approach where you would be able to visualize your subTADs. I understand these approaches cost can be higher yet if not that at least the addition of extra viewpoints inside and outside the subTADs would be needed to show the insulation experimentally. Although unlikely, what is shown here can simply be explained by the 4C interaction frequency dropping exponentially from the viewpoint that is conveniently located at the center of the subTAD.

We have addressed this issue by adding to the manuscript 4C analyses for a total of 12 virtual 4C viewpoints located at 6 intra-TAD loop anchors. These new data, which are presented in the third paragraph of the subsection “Intra-TAD loops show strong, directional interactions and insulate chromatin marks” and are shown in Figure 3B and Figure 3—figure supplement 1, are based on our analysis of the high resolution Hi-C data of Bonev et al., 2017, as proposed by the senior Editor. The viewpoints analyzed are both inside and outside of a predicted intra-TAD loop, just as the reviewer suggested, to further test the insulation of intra-TAD loop anchors directly. These interaction profiles presented provide evidence for the loop itself and also for its insulation of interactions, as the directionality of Hi-C interactions shifts when the viewpoint is moved from just outside to within the intra-TAD loop.

3) The extent of how the authors modified the method previously published and improved upon is largely unclear. I strongly suggest they make the code/scripts available to the community for all of their analyses otherwise their contributions most likely will remain unnoticed/unused.

All code and scripts have now been made available (Source code 1).

4) Concerns about novelty: besides the obvious similarity from approaches published before additionally there is a paper found on bioRxiv (Predicting CTCF-mediated chromatin interactions by integrating genomic and epigenomic features. Kai et al., 2017) that predicts subTADs based on machine learning of ChIA-PET results. Might be helpful to discuss the pros/cons between methods.

We added a section to the Discussion (fourth paragraph) where we highlight several notable differences between our approach and that described in the recent bioRxiv posting cited by the reviewer. Further, a key finding of their work, namely, that strength of cohesin binding is a more significant predictor of CTCF-mediated interactions than strength of CTCF binding, complements our work – where we introduce cohesin binding strength as a key component of our model – and supports the importance of considering cohesin strength in any TAD or subTAD/intra-TAD loop prediction model.

Reviewer #3:[…] 1) My one main concern is the missing out on an opportunity to clearly distinguish these subTADs from the contact domains defined by Rao et al., 2014 paper on several different cell lines. All the data needed to perform subTAD predictions are available for most (if not all) of these cell lines and a clear, cell-type matched comparison of what subTADs are and how they overlap with contact domains (as well as the subset of loop domains). Only in Figure S4C there is an indirect comparison of contact domains from a mouse B-cell lymphoma cell line (CH12-LX) and subTADs from mouse liver. I think it is crucial for the field to understand similarities and differences of many different definitions of domains and subdomains in order to converge to a consensus. This paper could have done more towards that goal.

We thank the reviewer for this constructive suggestion, as a converging consensus is critical to this rapidly expanding area of research. We have now applied our computational approach to identify intra-TAD loops in two mouse cell types (mESCs and NPCs) from Bonev et al., 2017, which provides CTCF/Cohesin ChIP, TADs, and sub-kilobase resolution Hi-C data (new data, results in Figure 2—figure supplement 3). We also applied our method for loop prediction to two human cell lines using data from Rao et al., 2014, which provides CTCF and Cohesin (Rad21) ChIP-seq data for at least 2 independent biological replicates for GM12878 and K562 cells. Results of these new findings have been added to the manuscript, and are shown in the subsection “Intra-TAD loops in other mouse tissues and in human cells”, and in Figure 2—figure supplements 3 and 4. These new data show that there is greater overlap of our predicted loops with loop domains than contact domains in the human cell lines, and that there is substantial overlap of CTCF-mediated ChIA-PET loops in both the mouse and the human models. Predicted intra-TAD loops are at least as tissue-conserved as TADs, and more so than loop/contact domains. Loops that are predicted across multiple cell types or by both ChIA-PET and our approach tend to be much stronger, suggesting that a substantial core of strong loop predictions are consistently identified across multiple approaches.